# Global Quantitative Proteomics Studies Revealed Tissue-Preferential Expression and Phosphorylation of Regulatory Proteins in *Arabidopsis*

**DOI:** 10.3390/ijms21176116

**Published:** 2020-08-25

**Authors:** Jianan Lu, Ying Fu, Mengyu Li, Shuangshuang Wang, Jingya Wang, Qi Yang, Juanying Ye, Xumin Zhang, Hong Ma, Fang Chang

**Affiliations:** 1State Key Laboratory of Genetic Engineering and Collaborative Innovation Center for Genetics and Development, Ministry of Education Key Laboratory of Biodiversity Sciences and Ecological Engineering and Institute of Biodiversity Sciences, Institute of Plant Biology, School of Life Sciences, Fudan University, Shanghai 200438, China; lovejianan@yeah.net (J.L.); fuying91@126.com (Y.F.); mengyulee@126.com (M.L.); flora_wang1004@163.com (S.W.); wangjingya1995@126.com (J.W.); qiyang18@fudan.edu.cn (Q.Y.); yejuanying@hotmail.com (J.Y.); xumin_zhang@fudan.edu.cn (X.Z.); 2Department of Biology, the Huck Institutes of the Life Sciences, The Pennsylvania State University, University Park, PA 16802, USA

**Keywords:** *Arabidopsis thaliana*, floral development, quantitative proteomics, quantitative phosphoproteomics

## Abstract

Organogenesis in plants occurs across all stages of the life cycle. Although previous studies have identified many genes as important for either vegetative or reproductive development at the RNA level, global information on translational and post-translational levels remains limited. In this study, six *Arabidopsis* stages/organs were analyzed using quantitative proteomics and phosphoproteomics, identifying 2187 non-redundant proteins and evidence for 1194 phosphoproteins. Compared to the expression observed in cauline leaves, the expression of 1445, 1644, and 1377 proteins showed greater than 1.5-fold alterations in stage 1–9 flowers, stage 10–12 flowers, and open flowers, respectively. Among these, 294 phosphoproteins with 472 phosphorylation sites were newly uncovered, including 275 phosphoproteins showing differential expression patterns, providing molecular markers and possible candidates for functional studies. Proteins encoded by genes preferentially expressed in anther (15), meiocyte (4), or pollen (15) were enriched in reproductive organs, and mutants of two anther-preferentially expressed proteins, *acos5* and *mee48*, showed obviously reduced male fertility with abnormally organized pollen exine. In addition, more phosphorylated proteins were identified in reproductive stages (1149) than in the vegetative organs (995). The floral organ-preferential phosphorylation of GRP17, CDC2/CDKA.1, and ATSK11 was confirmed with western blot analysis. Moreover, phosphorylation levels of CDPK6 and MAPK6 and their interacting proteins were elevated in reproductive tissues. Overall, our study yielded extensive data on protein expression and phosphorylation at six stages/organs and provides an important resource for future studies investigating the regulatory mechanisms governing plant development.

## 1. Introduction

Unlike animals, which complete most of their organ differentiation during embryonic development, plants undergo a continuous developmental process, with organogenesis occurring throughout the life cycle. With the two apical meristems that form during embryogenesis, plants develop the root system and the shoot [1]. The shoot grows above the ground against gravity to access light for photosynthesis, with responses to environmental cues, such as light, through light receptors, including both phytochromes (red and far-red light) and cryptochromes (blue light) [2,3,4,5]. In comparison, roots grow down in to the soil to take up nutrients and water in addition to achieving soil anchorage [6]. Postembryonic root growth relies on the combined action of transcription factors and mediators of intercellular signals, particularly the phytohormones auxin and cytokinin [7,8,9,10].

In addition to photosynthesis, leaves also perceive photoperiodic signals and then generate and transmit the florigen signal that regulates the transition to flowering at the shoot apex [11]. The transition from the vegetative to the reproductive phase involves various internal and external factors, such as photoperiod and temperature. During the reproductive process, floral organ primordia in several whorls develop into floral organs, including sepals, petals, stamens, and pistils, from outside to inside [12,13]. The development of *Arabidopsis* flowers from initiation until the bud opening is divided into 12 stages according to a series of landmark events [12]. During flower stages 1–9, the floral meristem initiates and expands, with subsequent formation of floral organ primordia followed by morphogenesis of the four types of floral organs; in addition, the microspore mother cells complete meiosis, resulting in tetrads of microspores that fill each anther lobe. In flower stages 10–12, the flower size increases rapidly, the gynoecium develops stigma and style, and microspores are released from the tetrads and develop into mature pollen with well-organized pollen exine. Finally, at flower stage 13, the sepals open, and anther dehiscence occurs to release the pollen grains, and the stigma is receptive [12,13].

Investigations in the last several decades have revealed multiple regulatory genes important for plant reproductive development [14,15]. Floral organs are specified by ABC(E) functional genes, including *AP1, AP2, AP3, PI, AG*, and the *SEP1/2/3/4* genes [13]. The stamen identity is determined by the combinatorial actions of BCE genes, including *AP3, PI, AG*, and *SEP1/2/3/4* [13,16,17]. Furthermore, anther morphogenesis is also regulated by receptor-like protein kinases (RLKs) including ER/ERL1/ERL2, BAM1/2, PRK2, EMS1, SERK1/2, and CIKs [18,19,20,21,22,23,24,25], cytosolic protein kinases including MAPK3/6 [26,27], and transcription factors including SPL, DYT1, AMS, MYB35/TDF1, and MYB103/80/MS188 [13,17,28,29,30,31,32,33,34,35,36]. In addition, a series of genes were demonstrated to be required for meiosis to produce microspores through the precise regulation of sister cohesion, pairing, synapsis, crossover, spindle organization, and chromosome segregation [13,17,37].

Phosphorylation is a protein modification catalyzed by kinases, whereas dephosphorylation is accomplished via phosphatases [38]. Phosphorylation is one of the most important protein modifications [39] and plays a crucial role in signal transduction by altering protein activities, protein interactions, or subcellular location [40,41]. Plant genomes encode approximately two times as many kinases as mammalian genomes [42], with 1052 annotated protein kinases in *Arabidopsis thaliana* as well as 162 phosphatases [43], strongly suggesting the important role played by protein phosphorylation in regulating cellular processes in the life cycle of plants.

Previous studies in plants have uncovered many phosphoproteins and their phosphorylation sites [44]. The phospho-proteome of organs (e.g., roots or shoots), cells, or subcellular components (e.g., plasma membranes) was investigated under different growth and culture conditions [45,46,47,48,49,50,51,52,53,54,55,56,57,58]. The data are available in public databases such as PhosPhAt [59] or Plant Protein Phosphorylation Database (P3DB) [60]. Another study was conducted to identify and analyze stamen proteins at stamen developmental stages 4–7 (near the time of meiosis) and 8–12 (microspore and pollen development) [61], albeit without quantitative comparison of expression levels of proteins in different tissues.

In this study, six stages/organs of the model plant *Arabidopsis thaliana* were studied: two from vegetative (16-day-old) plants (roots and above-ground parts (AGP)) and four from adult plants (cauline leaves, stage 1–9 flowers, stage 10–12 flowers, and open flowers). We performed high-accuracy mass spectrometry for iTRAQ-based quantitative proteomic and phosphoproteomic analyses. This study provides extensive data on the differences in protein levels and phosphoproteins between different plant stages/organs, enabling the formulation of hypotheses regarding protein functions and regulatory networks during plant development.

## 2. Results

### 2.1. Quantitative Identification of 2187 Proteins and 1194 Phosphoproteins in Six Stages/Organs

To identify proteins that likely function at various *Arabidopsis* developmental stages, we collected six developmental stages/organs, including roots and above-ground parts (AGPs containing hypocotyl, cotyledons, and leaves) of 16-day-old plants, cauline leaves (CL) from 6-week-old plants, stage 1–9 floral buds (F1–9) including floral meristem, stage 10–12 floral buds (F10–12), and open flowers (OF) (Figure 1A–C). All plants for tissue sampling were grown in the same growth chamber with the same growth conditions, and for each tissue/organ, the sample for either proteomics or phosphor-proteomics was collected from more than 100 plants to reduce the variation due to individual plant differences. In addition, three replicates for each tissue were obtained, and global profiling of the quantitative proteome and phospho-proteome of each sample was performed following the workflow indicated in Figure 1D.

In total, 11,341 unique peptides corresponding to 2187 non-redundant proteins were identified with an estimated FDR (false discovery rate) of 1% (Figure 2A, Appendix A), of which 2186 proteins with quantitative information were obtained. Pearson correlation was performed to determine the analytical reproducibility of the three replicates of each tissue (Appendix A), and the results showed that all values of the correlation coefficients were approximately 0.8, indicating good correlation among different replicates. 

In addition, quantitative phospho-protein analyses were performed for each of the six developmental stages/organs following the workflow indicated in Figure 1D. A total of 2497 unique phospho-peptides with 2893 detected phosphorylation sites in 1194 proteins were identified (Figure 2A, Appendix A). Among these proteins, 926 phospho-proteins with 1475 phosphorylation sites have been reported in the Plant Protein Phosphorylation Database (P3DB, http://www.p3db.org) (Figure 2B,C), suggesting that our data are relatively reliable. In addition, 268 phospho-proteins with 1022 phosphorylation sites were not included in the P3DB (Figure 2B,C), indicating the phosphorylation modification of these proteins are newly identified and further suggesting a great depth of our detection.

We analyzed the distribution of phospho-proteins with different numbers of phosphorylation sites and found 651 (55%) and 273 (23%) phospho-proteins with one and two phosphorylation sites, respectively; however, 270 phospho-proteins (23%) were detected to have three or more sites. Among the 2893 identified phosphorylation sites, 85.76% (2481), 12.34% (357), and 1.90% (55) were determined to target serine (pSer), threonine (pThr), and tyrosine (pTyr) amino acids, respectively (Figure 2D).

We also analyzed the times of detection for each identified peptide in total three replicates for both proteomic and phospho-proteomic analysis. In the proteomic data, 47%, 25%, and 28% of peptides were identified three times, two times, and once, respectively (Figure 2E). In addition, 40%, 28%, and 32% peptides were identified three times, two times, and once, respectively, in phospho-proteomic analysis (Figure 2F). Furthermore, we analyzed the Peptide-Spectrum Matches (PSMs) in total proteome peptides and phospho-peptides. The PSMs of most peptides varied from 1 to 50, and the PSMs of phospho-peptides ranged from 1 to 80 (Figure 2H).

To present the expression differences between the AGPs and roots and between the reproductive organs (including F1–9, F10–12, and OF) and the cauline leaves, we generated volcano plots for each pair in a comparison for proteomic and phospho-proteomic data. The differences in the expression levels of most peptides were more than 1.5-fold, with a *p*-value cutoff of 0.05 (Figure 3). Compared to those in roots, most of the detected peptides and phospho-peptides were increased significantly in AGPs (Figure 3A,B). Comparing to cauline leaves, approximately half of the detected peptides were up-regulated significantly in F1–9 (Figure 3C), F10–12 (Figure 3E), and OF (Figure 3G), and almost all phospho-peptides were up-regulated significantly in F1–9 (Figure 3D), F10–12 (Figure 3F), and OF (Figure 3H). These results suggest that the AGPs might require more proteins and phosphorylation modifications than roots, as do the reproductive organs/stages in comparison with the vegetative cauline leaves.

### 2.2. Sixteen-Day AGPs and Roots Have Distinct Protein Profiles

To investigate proteins that potentially have broad or distinctive functions in roots and/or AGPs, we first carried out protein expression comparisons between these two tissues. Interestingly, 1985 of the 2002 AGP-root differentially expressed proteins showed obviously increased expression in AGPs compared with roots (fold change of AGP/R, |FC| ≥ 1.5-fold), and only 17 proteins showed obviously reduced expression in AGPs (Figure 4A,B). These results suggest that many more proteins are required for development and physiology of AGP than that for roots. Among the 1985 AGP-preferential proteins, 369 showed a 1.5- to 2-fold increase, 940 showed a 2- to 4-fold increase, and 676 showed a greater than 4-fold increase (Figure 4B). Among the 17 root-preferential proteins, the fold change of seven proteins was over 4-fold (Figure 4B).

To investigate the functions of these differentially expressed proteins, we performed a gene ontology (GO) analysis of the 1985 AGP-preferential proteins and the 17 root-preferential proteins (Figure 4C–E). Most of the 1985 proteins were enriched in basic cellular processes, including photosynthesis (62 of the 84 enriched proteins showed ≥4-fold increased expression compared with roots, for short, 62/84), glucose metabolic processes (12/59), and the response to cadmium ions (38/182). In addition, proteins in abiotic stimulus processes (104/284), salt stress responses (32/113), and responses to cold (38/89) were enriched as well (Figure 4C,D). These results are consistent with previous findings that light intensity, CO2 concentration, water availability, and temperature have combined effects on proliferation, cell expansion, and endoreduplication and subsequently affect shoot organ growth [62]. The molecular function (MF) analysis also showed that many of the 1985 proteins were involved in GTPase activity, translation factor activity, rRNA binding, and similar processes (Figure 4E).

However, the 17 root-preferential proteins were clearly enriched in response to oxidative stress and hydrogen peroxide (Figure 4D). These 17 proteins included (Appendix A): ZCE2, which is reported to function in plant bolting [63]; a class III peroxidase member RCI3, which has shown peroxidase activity in root cells [64] and is involved in reactive oxygen species (ROS) production [65]; IAA-CONJUGATE-RESISTANT4 (IAR4), which is critical in root development, root hair formation, and auxin response [66]; and JAL34 and PLAT2.

### 2.3. Hierarchical Clustering Analysis of the Quantification of the Total Proteome

To identify possible correlations between protein expression level and plant development and putative organ/stage preferential functional proteins, we performed a cluster analysis (K_means = 15) on the 2186 proteins using RStudio and Euclidean distance classification. A total of 15 clusters with different expression patterns in the evaluated stages/organs were generated (Appendix A). Among them, 89 proteins were expressed at similar levels in all six stages/organs (Appendix A). Gene ontology (GO) analysis of these 89 proteins showed that the enriched GO categories are response to cadmium ions, oxidation-reduction processes, response to salt stress, tricarboxylic acid cycle, malate metabolic process, calcium ion transmembrane transport, and similar processes (Appendix A). In total, 17 proteins showed significantly preferential expression in 16-day-old roots (Appendix A) and were the same as the 17 root-preferential protein detected in the AGP/R comparison (Figure 4 and Appendix A). In addition, 12, 40, 72, and 134 proteins were found to show significantly preferential expression in 16-day-old AGP, 6-week-old CL, F1–9, and F10–12, respectively (Appendix A).

Interestingly, a total of 729 proteins showed obviously higher expression level in the three floral stages than in the other three tissues (Appendix A). The result that more proteins are relatively highly expressed in the reproductive organs/stages suggests that the normal reproductive process might need more complex functions of many proteins. In addition, proteins that were found to be preferentially expressed in certain organs/stages might be particularly important for the normal development and the physiology of the corresponding organ/stage.

To confirm the reliability of the quantitative proteomic data, especially for those protein clusters that showed some pattern(s) of stages/organ-preferential expression, we compared our proteomic data from six stages/organs to the transcriptomic data of the same or highly similar stages that are available in the eFP database (http://bar.utoronto.ca/efp/cgi-bin/efpWeb.cgi). As eFP has no transcriptomic data for stages exactly matching the 16-day AGP, stage 1–9 floral buds (F1–9), stage 10–12 floral buds (F10–12), and open flowers (OF) as those in our proteomic study, we used transcriptomic data from rosette leaves, flower 9, flower 10/11 and 12, and flower 15, respectively. The results of the comparison showed a relatively high degree of correlation, as illustrated by the following clusters: (1) for the 17 proteins that showed root-preferential proteomic expression pattern, 15 (88%) showed a consistent pattern at the transcript level (Figure 5A); (2) for the 40 cauline leaf (CL)-preferentially expressed proteins, transcript level of 40 (100%) in CL was considerably higher than that in roots and floral buds (Figure 5B); (3) for the 72 early floral buds (EF, flower 1–9)-preferentially expressed proteins, 37 (51%) exhibited the highest transcript level in stage 9 flowers (Figure 5C); (4) 43 of the genes encoding proteins that were identified as preferentially expressed in the late flower (LF, flower 10–12) were found to be preferentially expressed in flower 10/11 and/or flower 12 (Figure 5D). Overall, the stage/organ preferential expression in transcript level further supported the proteomic results in our study.

### 2.4. Distinct Expression Profiles of Proteins in Three Floral Stages Compared to Cauline Leaf

To further investigate proteins preferentially expressed in reproductive development, we first compared the expression levels of proteins among three different floral stages and the cauline leaf. Compared to the expression levels in cauline leaves, expression levels of 1766 proteins showed obvious changes (over 1.5 fold) in at least one of the three floral stages, including: 1445 proteins in stage 1–9 flowers (1303 increased and 142 decreased), 1644 proteins in stage 10–12 flowers (1598 increased and 46 decreased), and 1377 in open flowers (OF) (1291 increased and 86 decreased) (Figure 6A). Next, we compared the protein expression data of these three flower developmental stages to uncover common and distinctive changes. Expression of 1212 of the 1766 proteins was altered in all F1–9, F10–12, and OF stages, 125 were found in both F1–9 and F10–12 stages, 34 were found in both F1–9 and OF stages, and 117 were found in both F10–12 and OF (Figure 6B). In addition, 74, 190, and 14 proteins, respectively, were found higher in F1–9, F10–12, and OF than cauline leaves (Figure 6B). The 1212 proteins shared by all three floral stages were enriched in GO categories of both cellular processes and environmental responses. For cellular processes, ribosome biogenesis, ubiquitin-dependent protein catabolic process, formation of translation initiation complex, response to cytokinins, photosynthesis, fatty acid biosynthetic process, and microtubule-based processes were highly enriched. In addition, response to abiotic stresses including salt stress, cold, heat, and cadmium ion were strongly enriched (Figure 6C).

### 2.5. Anther-, Meiosis-, and Pollen-Preferentially Expressed Proteins Were Identified

Angiosperm male reproductive development includes anther morphogenesis, meiosis, and pollen development. Previous studies have identified many genes preferentially expressed during these processes, such as anther-preferential expressed genes [67], meiosis-preferential genes [68], and pollen genes [67]; however, expression and function of proteins encoded by most of these genes have not been investigated.

To identify proteins with putative functions in anther development, meiosis, and pollen development, we compared our quantitative proteomic data with previously published anther-preferentially [67], meiosis-related [68], and pollen-preferentially expressed [67] gene clusters from transcriptomic profiling (Figure 7A). We identified 15 anther-preferentially expressed genes that encode proteins among our proteomics results. Among the 15 proteins, 14 were more highly expressed in the three reproductive stages than in the cauline leaves, including 10 with the highest expression level in F10–12 (QRT3, KIN1, TKPR2, SCPL49, ACP1, AT3G23770, ACOS5, ATA27, MEE48/A6, and AT1G66850) (Figure 7B). The other four proteins, RGP3, AGO9, NUDT3, and ACLB-2, showed similar expression levels in all three flower stages (Figure 7B). Previous transcriptomic data showed that these genes were preferentially expressed in stage 4–7 anthers, which are part of stage 9 flowers [67]. In this study, our quantitative proteomics data indicated that their protein products might have been present for a considerably longer time and played roles even in the open flower when pollen development had completed and pollen grains were released from the anther. The high expression levels of these proteins during flower development strongly suggests their involvement in these processes. Previous studies based on phenotypic analyses suggest that QRT3 plays a direct role in degrading the pollen mother cell wall during microspore development [69], ACOS5 participates in a conserved biochemical pathway of sporopollenin monomer biosynthesis and is required for primexine formation [70,71], and MEE48/A6 is important for the dissolution of callose wall around the microspore tetrads [72,73]. In addition, our observation on another two T-DNA insertional mutant alleles, one *ACOS5* allele *acos5-4*, and one *MEE48* allele *mee48-2* (Figure 7C), also showed that, unlike the wild type plants that produced normal pollen grains with well-organized pollen exine structure, the *acos5-4* mutant had shriveled pollens with abnormal pollen exine structure. Pollen defects of *mee48-2* were stronger, most pollen grains were shrunken, and the pollen exine components were not even connected (Figure 7D).

On the other hand, one of the 15 anther-preferentially expressed genes, *BGLU44*, was shown to be preferentially expressed in stage 4–7 anthers, but the BGLU44 protein showed nearly equal levels in all examined organ/tissues, except for its lower expression in roots (Figure 5B), suggesting possible post-transcriptional regulation of BGLU44. In addition, proteins of four meiosis- and 15 pollen-related genes were identified, and most of them were more strongly expressed in the reproductive organs than in the vegetative tissues, except for MPA1 (Figure 7E,F).

### 2.6. Different Expression of Defense-Related Proteins in Different Organs

To identify the organ/stage-preferential expression of defense proteins, the 275 proteins that show clearly sage/organ-preferential expression pattern (Appendix A) were searched with keyword “defense”. Nine defense related proteins were identified (Appendix A). Among them, three proteins showed strong cauline leaf-preferential expression. These three proteins include EPITHIOSPECIFIER MODIFIER 1 (ESM1) that mediates indol-3-acetonitrile production from indol-3-ylmethyl glucosinolate for defenses against insect herbivores and various pathogens [74], CARBONIC ANHYDRASE 2 (CA2) that is structurally required for the assembly of Summary Complex I mitochondrial electron transport chain (mETC) and plays important role in reproductive development [75], and PHOTOSYSTEM II SUBUNIT P-1 (PSBP-1) that participates in the regulation of oxygen evolution and is involved in defense response to temperature and bacterium [76,77].

On the other hand, compared to their expression in vagetative tissues, the expressions of six proteins were clearly higher in the three flower developmental stages, especially F10–12 (Appendix A). These six proteins are EPITHIOSPECIFIER PROTEIN (ESP), TSK-ASSOCIATING PROTEIN 1 (TSA1), MYROSINASE-BINDING PROTEIN 1 (MBP1), MLP-LIKE PROTEIN 28 (MLP28), MLP-LIKE PROTEIN 423 (MLP423), and F9D16.150 (Appendix A). Among these proteins, ESP is suggested to be involved in pathogen resistance and leaf senescence [78,79,80]; TSA1 is suggested to be Jasmonic acid (JA) inducible and facilitates ERECTA (ER) body formation [81] and is involved in nuclear architecture [82] and seedling development in darkness [83].

### 2.7. Quantitative Phosphoproteomics Overview

Protein phosphorylation is one of the most important post-translational modifications regulating plant cellular functions. To investigate global protein phosphorylation patterns in major *Arabidopsis* organs/stages, we compared the phosphorylation levels of proteins among the six examined plant organs/stages, using quantitative proteomic analyses. To identify certain tissue-specific protein phosphorylation, we searched the phosphor-proteomic data for peptides being fully absent in one or more organs/stages. Among the 2497 detected phospho-peptides with 2893 detected phosphorylation sites in 1194 proteins, 51 peptides with 83 detected phosphorylation sites in 45 proteins were absent in root (R) but present in the other five organs/stages (AGP, CL, F1–9, F10–12, OF); two peptides with two detected phosphorylation sites in two proteins were absent in above-ground-parts (AGP) but present in the other five organs/stages (R, CL, F1–9, F10–12, OF); and two peptides with two detected phosphorylation sites in two proteins were absent in cauline leaves (CL) but present in the other five organs/stages (R, AGP, F1–9, F10–12, OF); another 2442 (97.8%) phospho-peptides with 2806 (97.0%) phosphorylation sites in 1145 (95.9%) proteins were present in all the six examined organs/stages (Appendix A). These results suggest that most of the phosphorylation regulation likely exhibited a more-or-less but not yes-or-no differences among different organs/stages. Therefore, we further investigated the organ/stage-preferential phosphorylation. Between AGPs and root, we identified 995 phospho-proteins with 1956 phosphorylation sites that were differentially phosphorylated (Appendix A). Similar to the previously mentioned greater number of proteins showing increased expression in AGPs, most of the phosphoproteins and the phosphorylation sites exhibited higher levels in AGPs than that in roots. In addition, 1149 proteins with 2313 phosphorylation sites showed different phosphorylation levels between OF and CL, with most of them showing obviously increased levels in OF (Appendix A). These results further demonstrated regulation of protein function via phosphorylation likely plays a greater role in the AGPs than that in roots, and the normal development and physiology of reproductive stages/organs need functions of more protein and phosphorylation modification.

To uncover the potential biological processes in which these phosphoproteins are involved, we then examined predicted subcellular localizations of the differentially phosphorylated proteins between flower and leaf (OF/CL;1149) and between AGP and root (AGP/R; 995) using a public database (WoLF PSORT, http://www.genscript.com/wolf-psort.html). As shown in Appendix A, 701 (61.0%) of the OF/CL differentially phosphorylated proteins and 603 (60.6%) of the AGP/root differentially phosphorylated proteins were predicted to localize to the nucleus (Appendix A), suggesting an important role of phosphorylation for nuclear-localized proteins, including transcription factors (TFs), in both vegetative and reproductive tissues. In addition, the numbers of phospho-proteins were similar in both the OF/CL and the AGP/R comparisons for localization to several subcellular locations or structures, including cytosol, cytoskeleton, extracellular space, mitochondria, chloroplast, and various membranes.

To obtain clues about regulation of TFs, we compared protein phosphorylation levels between the three floral developmental stages (F1–9 including floral meristem, F10–12, open flower), and the cauline leaves and examined the identified phosphoproteins for members of known TS families and their difference in phosphorylation levels. Among the 2296 *Arabidopsis* TFs (1717 loci) in 58 families in the PlantTFDB v4.0 (http://planttfdb.cbi.pku.edu.cn/), 62 TFs belonging to 22 families were identified here as phosphoproteins, including two ARF, three bHLH, six bZIP, eight C2H2, five GeBP, six MYB, two WRKY, and five ZF-HD family transcription factors (Appendix A). Furthermore, the numbers of identified phosphoproteins in each TF family were the same across all three floral stages, except those in the Trihelix family (Appendix A), suggesting that the extent of TF phosphorylation in the three floral developmental stages was similar.

We further compared the phosphorylation levels of the members of each TF family between F10–12 and F1–9 (F10–12/F1–9) and between OF and F10–12 (OF/F10–12). The results showed that only 11 TFs were differentially phosphorylated in F10–12/F1–9, including one bZIP, two C2H2, one C3H, two HB-other, one WRKY, and five ZF-HD family members, whereas 28 TFs belonging to 17 TF families were differentially phosphorylated in OF/F10–12, including the two ARF, two bZIP, three C2H2, one WRKY, and four ZF-HD family members (Appendix A). These results suggest that the phosphorylation modifications of TFs involved in developmental regulation between F1–9 and F10–12 are more similar than those between OF and F10–12. Moreover, 27 of the TFs showed decreased phosphorylation levels in F10–12/F1–9, but only one, ARF family member ARF8, showed an increased phosphorylation level. In OF/F10–12, 10 TFs showed decreased phosphorylation, and one ZF-HD family member, MIF2, increased (Appendix A).

### 2.8. Phosphorylation of Anther-, Meiosis-, and Pollen-Preferential Proteins

To uncover putative regulation by phosphorylation of protein encoded by previously reported anther-, meiosis-, and pollen-preferential genes, we compared our quantitative phosphorylation data to the anther-preferentially [67], meiosis-related [68], and pollen-preferentially expressed [67] gene clusters as well. Among all the detected phosphoproteins, one anther-preferential, three meiosis-related, and nine pollen-related gene encoded proteins were identified.

PolyA binding protein 5 (PAB5) was suggested to be anther-preferentially expressed, and in this study, we found a novel phosphorylation site in the second serine of the LASDLALpSPDK peptide of the PAB5 protein (Table 1). The phosphorylation level of this serine was higher in stage 1–9 floral buds than that in cauline leaves (Log2FC(F1–9/CL) = 1.58). Later in flower development, its phosphorylation became higher in F10–12 and significantly decreased in OF (Table 1). The phosphorylation level of the peptides from the meiosis-preferential protein MEI2-LIKE PROTEIN 5 (AML5) in F1–9 was higher than that in cauline leaves as well, then its phosphorylation decreased in F10–12 and OF. In comparison, two other meiosis-preferential proteins, SISTER-CROMATED COHESION PROTEIN 3 (SCC3) and ATM, showed dramatic increase in their phosphorylation levels in F1–9 compared with CL, and their phosphorylation levels remained high at later floral developmental stages (Table 1). As meiosis occurs at anther stage 6 in flower stage 9, the relatively higher levels of phosphorylation modification level of the meiosis-related genes in F1–9 were consistent with their transcriptomic expression pattern and functional stage.

Nine phospho-proteins related to pollen were also identified, and their phosphorylation levels were up-regulated in F1–9 related to CL, with six out of these nine proteins having only one detected phosphorylation site and three proteins with 3-to-5 phosphorylation sites (Figure 8A and Table 1). For instance, five, three, and four phospho-sites were identified in GLYCINE RICH PROTEIN 17 (GRP17), At4G28000, and At1G30470, respectively (Figure 8A and Table 1). GRP17 encodes a glycine-rich protein and is expressed specifically during flower stages 10 to 12 [84], At4G28000 is involved in nematode-induced syncytium development and abiotic stress responses [85], and At1G30470 is a phosphoatase-associated family protein [86]. Remarkably, the phosphorylation levels of the peptide H**pTpS**GNDLHSR of At4G28000.1 protein were extremely high in the three floral stages with similiar phosphorylation levels in F1–9 and F10–12 (Log2FC(F1–9/CL) = 3.45, Log2FC(F10–12/F1–9) = 0.00) and even higher phosphorylation level in OF (Log2FC(OF/F10–12) = 3.45), suggesting potentially important function of this phosphorylation in flower and pollen development. In addition, the phosphorylation level of the detected serine site in the CMSGGM**pS**GSEGGMSR peptide of GRP17 showed progressive increases in F1–9 and F10–12 but dramatic reduction at OF stage, although its phosphorylation levels in these three floral stages were all obviously higher than those in the CL (Figure 8A and Table 1).

In addition, several proteins encoded by pollen-related genes based on transcriptomic data [67] showed extremely high phosphorylated level in at least one floral organ. One of these proteins is AT1G52680.1, which is a late embryogenesis abundant protein-related/ late embriogenesis abundant (LEA) protein-like protein that is reported to be involved in flower development, pollen germination, and tube growth in *Arabidopsis* [87,88]. The phosphorylation level of the identified threonine (T) site in the NTLGMSPATNSPSSPAG**pT**TR peptide was much higher in the three floral stages than that in CL (Log2FC(F1–9/CL) = 2.41), remained the same level in F1–9 and F10–12, and increased to its highest level in OF (Log2FC(OF/ F10–12) = 1.37), suggesting a possible role for phosphorylation of this protein in flower developmental stages, especially in the pollen (Figure 8A and Table 1). Another one (AT4G28000.1) is a member of the P-loop containing nucleoside triphosphate hydrolases superfamily. Three phosphorylation sites were detected, and two of them showed progressive increase in phosphorylation level during flower development and reached its highest phosphorylation level in OF (Figure 8A and Table 1). Moreover, a member (AT1G30470.3) of the SIT4 phosphotase-associated family had four identified phosphorylation sites, and each of them showed highly reproductive-preferential phosphorylation pattern (Log2FC(F1–9/CL) = 3.63, 2.74, 2.7, 2.4, respectively), with three sites showing equal phosphorylation level in the three floral stages and one site showing obviously F1–9 abundant phosphorylation pattern (Figure 8A and Table 1).

### 2.9. Protein Kinases and Phosphatases Are More Likely to Be Phosphorylated in Reproductive Tissues

Protein kinases and phosphatases are often themselves regulated by phosphorylation. To obtain clues about possible functions of kinases and phosphatases in floral development, we compared protein and phosphorylation levels of kinases and phosphatases in the three floral stages, F1–9, F10–12, and OF, to those of the CL (Table 2 and Table 3). Five kinases of four families were identified from the quantitative proteomics analysis, among which four kinases of CDPK, LRR_3, and SnRK2 families, CDPK6, RLK1, RLK902, and SNRK2-10, were up-regulated in floral stages (Table 2).

In addition, 27 kinases/phosphatases belonging to 15 families showed obviously enhanced phosphorylation levels in at least one of the floral stages, as indicated by the quantitative phosphoproteomics results (Table 3). These results suggest that the functions of kinases and phosphatases in the floral stages are likely regulated by phosphorylation. The phosphorylated kinases include three MAP3K proteins, one MAP2K protein, three CDK proteins, four CDPK proteins, and four LRR proteins (Figure 8B and Table 3). Among the phosphorylated kinases are the receptor kinase ERECTA (ER) and MPK6; previously ER and its closely related ER-like 1 and ER-like 2 (ERL1 and ERL2) together were shown to be important for early anther development, as were MPK6 and its close paralog MPK3 [22]. These results suggest that the functions of ER and MPK6 in anther development are regulated by phosphorylation.

Furthermore, we compared their phosphorylation levels of the kinases and the phosphatases among the three floral stages with F10–12/F1–9 and OF/F10–12 (Table 4). In total, 12 proteins showed changed phosphorylation level during flower development, including nine kinases with reduced levels in F10–12 as compared to those in F1–9 and three proteins (two kinases and one phosphatase) showing reduced phosphorylation in the OF related to F10–12. Notably, two phosphorylation sites were detected in the peptides of MPK6, ATSK11, STY8 (STY8-3), and MPK16 (MPK16-2), respectively (Table 3), and both the phosphorylation level and the change pattern of the two sites in each peptide during floral development were found to be the same. These findings of the 27 floral-preferential phospho-proteins suggest important functions and regulation by phosphorylation in flower development. Moreover, almost half (12/27) of them showed much higher phosphorylation levels in F1–9, suggesting that early floral development likely involves more protein phosphorylation compared with later stages.

To further test for changes of the level of phosphoproteins, we scanned the entire list of detected phospho-proteins here for ones that have available antibodies; we found such antibodies for three detected phosphoproteins, GRP17, CDC2/CDKA.1, and ATSK11. We obtained the commercially available antibodies for western blot analysis. As shown in Figure 8C, only one GRP17 protein band was detected by the anti-GRP17 antibody in the protein gel in each examined organ/stage, whereas in the phos-tag gel, another band was present in both F1–9 and F10–12 stages, with relatively weak and much stronger signals, respectively (Figure 8C). This result confirmed the phosphorylation of GRP17 and its F10–12-preferential phosphorylation pattern (Figure 8A and Table 1). In addition, similar analyses of CDC2/CDKA.1 and ATSK11 also uncovered bands of reduced mobility of phospho-proteins in flower organs but not in cauline leaves and rosette leaves, further validating the quantitative phospho-proteomics results.

### 2.10. Phosphorylation Regulatory Network during Floral Development

To investigate the possible regulatory network mediated by protein phosphorylation, we predicted kinase-substrate relationships according to the results from motif analysis and the PhosPhAt database [42]. The analysis supported 71 predicted kinase-substrate pairs and their interaction types from preferentially phosphorylated proteins from F1–9/CL, 22 in F10–12/F1–9, and two in OF/F10–12 comparisons (Appendix A). We found that the putative substrates of MAPK6 were highly enriched in both F1–9/CL and F10–12/F1–9 comparisons, as part of the interaction network illustrated using Cytoscape 3.4.0 (Figure 9). Our results are consistent with the previous conclusion that MAPK6 plays an important role in flowering, especially in anther development [22]. Among the putative substrates of MAPK, one transcription factor, bZIP16, was identified as differentially expressed during floral development. bZIP16 has demonstrated that it can integrate light and hormone signaling pathways to regulate early seedling development and can heterodimerize with other G group members [89]. Our phosphorylation results on bZIP16 suggest an important role of phosphorylation in regulating bZIP16 activity during floral development.

Among other possible interactive kinases and substrates, components of the JA and the Abscisic Acid (ABA) signal regulatory network were up-regulated in flowers 1–9 when compared with the leaf. We also identified BSK1, which is one of three homologous Brassinosteroid (BR)-signaling kinases belonging to the RLCK_2 family and is autophosphorylated as part of BR signaling and ROS pathways [90]. Another identified kinase is KIN10, an SnRK1 family kinase, which is autophosphorylated during calcium signaling of plant growth and phosphorylates TPS5 and NIA2 during sugar signaling. A member of the *Arabidopsis* CDPK gene family, CDPK6, also had target proteins overrepresented in our network during floral stages 1–9 and 10–12, suggesting that CDPK6 may play important roles in floral development. The substrates of YDA and the LRR STY8 were also significantly identified in comparisons of both floral stages 10–12 with 1–9 and floral 1–9 to leaf.

## 3. Discussion

In flowering plants, vegetative processes and reproductive development both involve crucial pathways that directly determine the growth and other functions of the plant. Using iTRAQ-based quantitative proteomic and phosphoproteomic analyses, this study provides an overview of dynamic protein expression and phosphorylation during six *Arabidopsis* developmental stages at a whole-genome scale. In addition, comparative analyses of protein expression and phosphorylation modification reveal the distinct regulation of gene function at both translational and post-translational levels between vegetative and reproductive developmental processes and between different floral developmental stages.

Our data reveal that the photosynthetic process was enriched in rapidly growing organs, AGP in vegetative processes, and flowers in reproductive processes. More detailed analyses showed that most of the proteins related to photosynthesis were involved in light reactions, harvesting, and stimulus responses. Chlorophyll A/B binding proteins such as CAB3, LHCA1/2/3, and LHCB2/3/6 were identified in our differentially expressed proteins (DEPs) with stronger gene expression levels in leaves than in any other organs or tissues [91,92,93]. However, there are many differences between the vegetative and the reproductive phases. Translation and metabolic processes are very active in reproductive organs, whereas responses to different stresses and oxidation-reduction are more significant in seedlings. From the perspective of phospho-proteomics analysis, there are more phosphorylation events during reproductive development than in the vegetative process. The phospho-proteins with changes in phosphorylation levels showed more distribution in the nucleus during the reproductive process. Transcription factors (TFs) are among the most important elements in the nucleus. There is only one difference in the distribution of TF families between the two processes, namely, the expression of the Trihelix families.

Our proteomics and phospho-proteomics analyses focused on comparisons between vegetative and reproductive processes and among the reproductive phases from floral stages 1–9 to open flowers. Hundreds of phospho-proteins were found to be phosphorylated and dephosphorylated during flowering, especially transcription factors and kinases, which suggests their possible molecular mechanisms in regulating floral development. There were 27 transcription factors phosphorylated in the early floral stages compared to vegetative organs but dephosphorylated in floral stages 10–12. Another 10 TFs were dephosphorylated after flowers opened. In addition to transcription factors, kinases play significant roles in these processes. Nearly one third of the kinases showing high phosphorylation levels in floral stages 1–9 were dephosphorylated in stages 10–12. The phosphorylation networks between kinases and their targets also indicated that there are multiple hormone signaling pathways involved in the flowering process, including JA, ABA, BR, and ROS.

Proteomics has been widely used to generate comprehensive and quantitative map. Meanwhile, phospho-proteomics provides insights into the dynamic regulation of proteins by post-translational modifications (PTMs), which are critical for protein function. However, some questions resulting from iTRAQ-based proteomics remain to be resolved. Although our iTRAQ-based data provided protein abundance information for different organs, the proportions of specific proteins in various tissues are still unknown. Other questions are how the threshold value of differentially expressed proteins can be standardized, and when it is appropriate to use 1.2, 1.5, or 2. The use of a lower ratio such as 1.2 would include more quantification variations; however, there would be more false positive variations. In comparison, a higher ratio such as 2 might enhance the reliability of detected variations, but it also could possibly overlook useful information with small changes. Therefore, the key issue of selection of candidates should base on two aspects; one is the confidence level of the quantitative data, the correlation of biological triplicate, for example, and the other is how willing the scientist is to spend extra effort to validate the selected candidates with an alternative method including immunoblot, multiple reaction monitoring (MRM), or selected reaction monitoring (SRM) of samples.

Thus, iTRAQ-based proteomics generates relative expression levels, but it has a number of disadvantages. For instance, the iTRAQ method provides both a lower number and a lower percentage of differentially abundant proteins than the label-free method. At present, label-free quantitative methods are becoming more popular in proteomics and other biological studies. However, when time on the instrument or cost are important considerations, iTRAQ may be the method of choice. Increasingly, efforts have been made to improve proteome sample preparation and mass spectrometry instruments. In the future, a greater number of scientific issues will be investigated using proteomics and phospho-proteomics, allowing the identification of key genes, signaling pathways, or regulatory mechanisms.

## 4. Materials and Methods

### 4.1. Plant Materials and Experiment Design

*Arabidopsis thaliana* plants of the Columbia ecotype (Col-0) were used in this study. Seeds were surface-sterilized with 75% EtOH for 3 min, rinsed 3 times with sterile water, plated on plates containing half-strengthen Murashige and Skoog (1/2 MS) medium (Wako) with 1.5% agar and 1% sucrose (pH 5.8), and then stratified at 4 °C in the dark for 2 days. Plants were germinated and grown under long day conditions (16 h light/8 h dark; light intensity: 120 µmol m^−2^ s^−1^; humidity: 60–70%) at 22 °C until the roots (R) and the above-ground parts (AGP) of 16-day plants were harvested. The 16-day-old Col-0 plants were transferred to the soil and grown for another 4 weeks under the same light and temperature conditions, then cauline leaves (CL), stage 1 to 9 flowers (F1–9), stage 10 to 12 flowers (F10–12), and open flowers (OF) were collected and immediately frozen in liquid nitrogen, respectively, and stored at −80 °C until further analyses. For each tissue/organ in this study, the sample for either proteomics or phospho-proteomics was collected from more than 100 plants.

### 4.2. Protein Extraction and Peptide Preparation

Two grams of tissues from each stage/organ were ground to fine powder in liquid nitrogen and suspended with buffer containing 6 M Urea, 2 M Thiourea, and 100 mM NH4HCO3 (final concentration), respectively. Suspensions were sonicated for 60 min (2 s sonication with 5 s intervals) on ice, and the extracted proteins were collected by centrifugation at 20,000× *g* for 20 min at 4 °C. Then, proteins were reduced with 10 mM DL-Dithiothreitol (DTT) for 1 h at 37 °C and alkylated with 30 mM iodoacetamide for 1 h at room temperature (25 °C) in the dark. The excess iodoacetamide was quenched by the addition of 10 mM DTT. After another centrifugation at 30,000× *g* for 20 min at 4 °C, the protein concentration was determined using Quick Start Bradford Protein assay (Biorad, Hercules, CA, USA).

An aliquot of total protein (200 μg) was taken from each sample and digested with endoproteinase Lys-C at the ratio of 1:100 (Lys-C: protein) for 4 h at room temperature, followed by digestion with Trypsin (Promega, Madison, WI, USA) at the ratio of 20:1 (protein: trypsin) at 37 °C for 12 h.

### 4.3. iTRAQ Labeling and Desalting

After trypsin digestion, the peptides were dried by vacuum centrifugation, and 100 μg peptides for each tissue were processed according to the manufacturer’s protocol for 8-plex iTRAQ (Applied Biosystems). The six samples were labeled with the six iTRAQ-tags: 115-iTRAQ tag for F1–9, 116 for F10–12, 117 for OF, 118 for R, 119 for AGP, and 121 for CL.

Then, the iTRAQ-labeled peptides were mixed together, and 100 μg peptide mixtures were acidified with Formic Acid (FA) and loaded onto a pre-equilibrated homemade Poros R3 microcolumn. After washing the R3 resin twice with 5% FA, the peptides were eluted by 20 μL of 30% Acetonitrile (ACN), followed by 20 μL of 60% ACN.

### 4.4. Nanoflow LC-ESI-MS/MS Analysis Based on Linear Trap Quadrupole (LTQ)-Orbitrap Elite

Liquid chromatography (LC)-electrospray ionization (ESI) tandem MS (MS/MS) analysis was performed using a nanoflow EASY-nLC 1000 system coupled to an LTQ-Orbitrap Elite mass spectrometer (Thermo Fisher Scientific) following previously reported procedures (Zhang et al., 2015). A two-column system was adopted for all analyses. Each peptide fraction was first loaded onto an Acclaim prepMap100 C18 Nano Trap Column and then analyzed on an Acclaim Prep Map RSLC C18 Column. The peptides were eluted using the following gradients: 5–35% solution B (0.1% FA in 95% ACN) in 58 min, 35–90% B in 5 min, 90% B for 13 min at a flow rate of 200 nL/min.

The MS/MS mode was set as follows: activation type, higher collision energy dissociation (HCD); collision energy, 35 eV; resolution, 60,000 FWHM; scan range, 300–1800 m/z; The peptide ions were detected in the Orbitrap mass spectrometer, and up to 15 of the most intense peptide ions (>5000 counts) were selected and fragmented by MS/MS. Information on peptides and peptide fragments *m*/*z* was collected as follows: 15 fragment spectra were collected after every full scan (MS2 scan), HCD fragmentation, full scan at a resolution of 60,000, and normalized collision energy of 35 eV. Three technical replicates were conducted to ensure reliable statistical consistency.

### 4.5. Proteomic Data Analysis

Raw data files acquired from the LTQ-Orbitrap Elite were converted into MGF files using Proteome Discoverer 1.4 (Thermo Fisher Scientific, Germany), and the MGF files were queried against the *Arabidopsis* database (ftp://ftp.arabidopsis.org/home/tair/Sequences/blast_datasets/TAIR10_blastsets/TAIR10_pep_20110103_representative_gene_model_updated) using an in-house Mascot search engine 2.3.02 (Matrix Science, London, UK).

Data were searched using the following parameters: 10 ppm mass tolerance for MS and 0.05 Da for MS/MS fragmented ions, with an allowance for only tryptic peptides with up to two missed cleavage sites allowed. Only peptides with a significance score greater than 99% confidence interval by a Mascot probability analysis were counted as identified. The quantitative protein ratios were weighted and normalized by the Mascot search engine (Matrix Science), and only ratios with *p* < 0.01, as determined according to a Student’s t-test, were employed.

The summation of six labeled sample mixtures was used as background (BC) based on the weighted average of the intensity of report ions in each identified peptide. The relative expression of protein was normalized by the ratio of value in certain samples to that of the summation (F1–9/BC, F10–12/BC, OF/BC, R/BC, AGP/BC, CL/BC). Proteins that were once identified in three technical replicates were all considered. In the vegetative process, the final ratio of protein was obtained from the comparison AGP/BC to CL/BC. At the reproductive stage, the ratios of CL/BC were used as reference (REF), and all the three flower organs, F1–9, F10–12, and OF, were compared to REF, among which ratios greater than ±1.5-fold were considered to be differentially expressed proteins (DEPs).

### 4.6. Phosphopeptide Enrichment

In addition to total proteome quantification, most of the iTRAQ-labeled peptide mixtures (400 ug for each) were used for phosphopeptide enrichment using TiO2 microcolumns (1350L250W046 Titansphere, 5 mm, 250 4.6 mm, GL sciences Inc) as described by Thingholm et al. [94]. The mixtures were vacuum dried and resuspended with TiO2 loading buffer (1 M glycolic acid in 80% ACN, 5% Trifluoroacetic Acid (TFA)) and applied onto the TiO2 microcolumn. After washing four times with 20 μL loading buffer and at least three times with 20 μL washing buffer (80% ACN, 1% TFA), the bound peptides were eluted twice with 20 μL elution buffer 1 (2 M NH_3_•H2O) and with 2 μL elution buffer 2 (1 M NH3•H2O in 30% ACN). The eluates from TiO2 enrichment were desalted as described in the total proteome.

### 4.7. Phosphoproteomic Data Analysis

As with the proteome data, raw LC-MS/MS files from phoshoproteomics experiments were queried against the *Arabidopsis* database using an in-house processed using Mascot search engine 2.3.02 (Matrix Science, London, UK), with the same parameters except for four significant points: 1. phospho_STY (serine, threonine, and tyrosine) was added in the specified parameters in the protein database searches; 2. identified peptides were further validated with a Target Decoy PSM validator, and phosphorylation sites were evaluated with phosphoRS3.0; 3. a false discovery rate (FDR) of 1.0% was used for the identification of phosphorylation residues (phosphosites); 4. regarding the phosphorylation levels of phosphoproteins, we determined the ratios of each phosphosites manually instead of using the calculated results from the Proteome Discover.

### 4.8. Bioinformatic Analysis

Gene ontology (GO) analysis of differentially expression genes was performed using online DAVID analysis (http://david.abcc.ncifcrf.gov/). Pathway enrichment analysis was performed using the list of genes that encode for proteins containing the peptide/phosphopeptides from each of the inferred cluster. Pathway enrichment with a set of genes was evaluated by comparing that set of genes against genes within known pathways using Mapman software. The subcellular localization of proteins and phosphoproteins was predicted with WoLF PSORT (http://above-ground.genscript.com/wolf-psort.html). R language (x64 3.3.1) was used to cluster the DEPs and to draw heatmaps. The relationship of detected kinases and potential substrates was generated through PhosPhATDB (http://phosphat.uni-hohenheim.de/) [42].

## Figures and Tables

**Figure 1 ijms-21-06116-f001:**
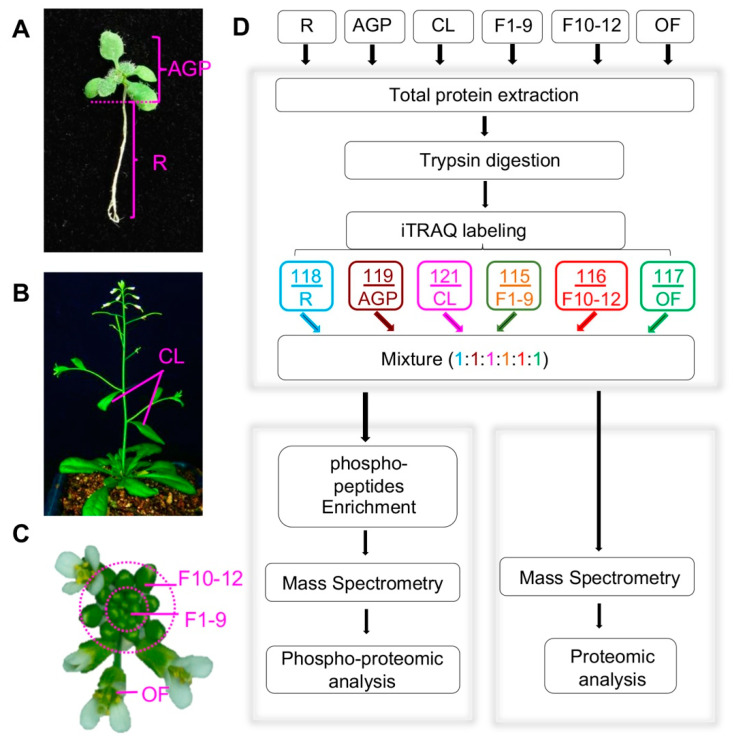
Plant tissue collection and the procedure for quantitative proteomics and phospho-proteomics analyses. (**A**) Above-ground parts (AGP) and roots (R) of 16-day-old plants. (**B**) Front view of plants during the flowering period, including cauline leaves (CL) and inflorescences (Flower). (**C**) Col-0 inflorescence including stage 1–9 flowers (F1–9), stage 10–12 flowers (F10–12), and open flowers (OF). (**D**) Workflow of our iTRAQ-based quantitative proteomics and phospho-proteomics analyses. The arrows represent experiment process.

**Figure 2 ijms-21-06116-f002:**
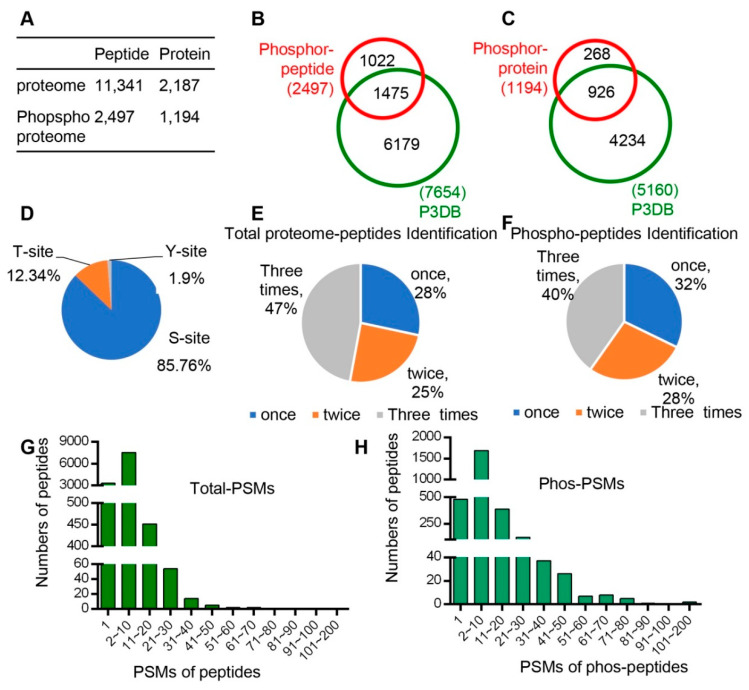
All peptide and phsophopeptide data were summarized and compared with database data. (**A**) Number of peptides and unique proteins that we identified from proteomes and phospho-proteomes. (**B**) Comparison of phospho-pepitides and the P3DB database (the Plant Protein Phosphorylation Database, http://www.p3db.org). (**C**) Comparison of phospho-proteins and the P3DB database. (**D**) Distribution of phosphorylated amino acids (S, T, Y) in the phospho-proteome. (**E**) Distribution of times of identification in total proteome peptides. (**F**) Distribution of times of identification in phospho-peptides. (**G**) Distribution of Peptide-Spectrum Matches (PSMs) in total proteome peptides. (**H**) Distribution of Peptide-Spectrum Matches (PSMs) in phospho-peptides.

**Figure 3 ijms-21-06116-f003:**
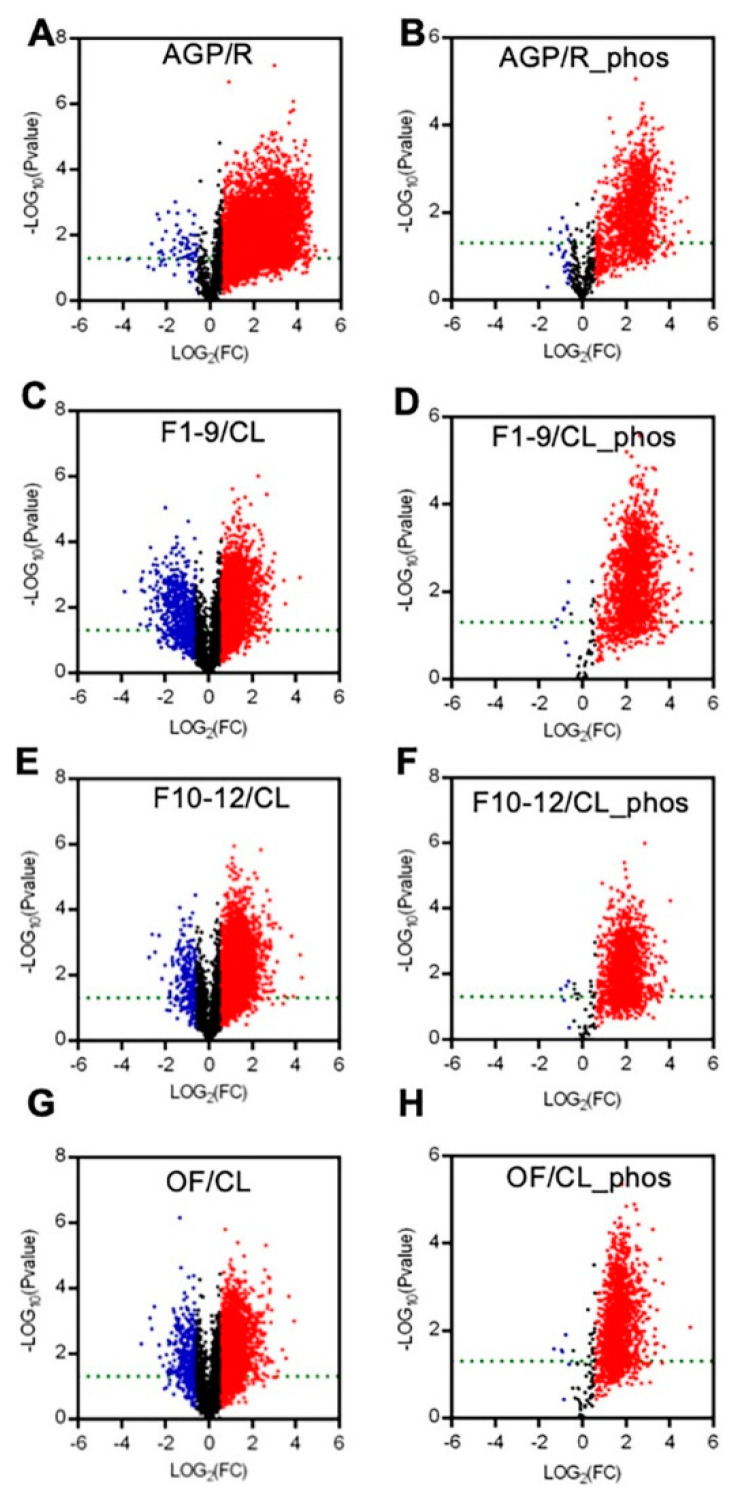
Summary of the expression levels of differences between AGP and R, F1–9 and CL, F10–12 and CL, OF and CL in quantitative peptides and phosphopeptides. (**A**) Volcano plot between R and AGP peptide samples. (**B**) Volcano plot between R and AGP phosphopeptide samples. (**C**) Volcano plot between F1–9 and CL peptide samples. (**D**) Volcano plot between F1–9 and CL phosphopeptide samples. (**E**) Volcano plot between F10–12 and CL peptide samples. (**F**) Volcano plot between F10–12 and CL phosphopeptide samples. (**G**) Volcano plot between OF and CL phosphopeptide samples. (**H**) Volcano plot between OF and CL phosphopeptide samples. R, 14-day-old root; AGP, 14-day-old tissues above ground part; CL, cauline leaves; F1–9, stage 1–9 flowers; F10–12, stage 10–12 flowers; OF, opened flowers. The red denotes log2FC > 1 and *p*-value < 0.05. The blue denotes log2FC < −1 and *p*-value < 0.05. The black denotes −1 < log2FC < 1 or *p*-value > 0.05.

**Figure 4 ijms-21-06116-f004:**
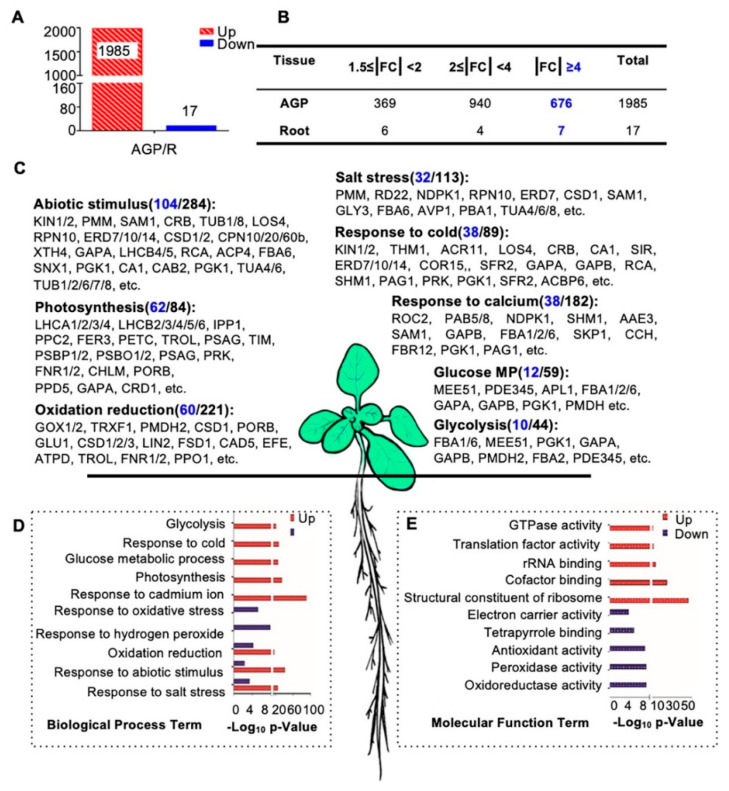
Biological process and molecular function enrichment analysis of AGP-root differentially expressed proteins. (**A**) Numbers of differentially expressed proteins in the comparison of above-ground parts (AGP) and roots in 16-day-old plants: 1985 proteins up-regulated and 17 proteins were downregulated in aboveground parts. (**B**) Detailed distribution of differentially expressed proteins. The highlighted number denotes numbers of protein with |FC| ≥ 4. (**C**) Biological process enrichment of upregulated proteins in the AGP. The highlighted number denotes numbers of protein with |FC| ≥ 4. (**D**) Biological process enrichment of differentially expressed proteins in AGP. (**E**) Molecular function enrichment of differentially expressed proteins in the aboveground parts. Counts in blue indicate the number of proteins with fold changes greater than four. The *p*-value is the enrichment score (E-score) from DAVID (http://david.abcc.ncifcrf.gov/). Higher-log10 p-values indicate more significant enrichment.

**Figure 5 ijms-21-06116-f005:**
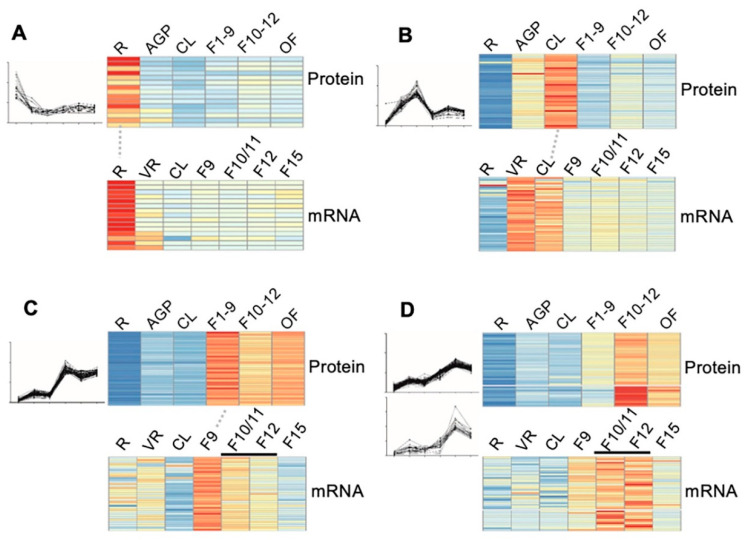
Correlation of the dynamic expression pattern between the protein level and the mRNA level. (**A**–**D**) The expression of the 17 root preferentially expressed (**A**), the 40 cauline leave (CL)-preferentially expressed (**B**), the 37 stage 1–9 flower (F1–9) preferentially expressed (**C**), and 43 stage 10–12 flower (F10–12) preferentially expressed (**D**) proteins and their correlated transcriptomic data from the eFP database (http://bar.utoronto.ca/efp/cgi-bin/efpWeb.cgi). R, 14-day-old roots; AGP, 14-day-old above ground part tissues; CL, cauline leaves; F1–9, stage 1–9 flower buds; F10–12, stage 10–12 flower buds; OF, opened flowers; VR, vegetative tissue; F10/F11, stage 10 or stage 11 flowers; F12, stage 12 flowers; F15, stage 15 flowers. The red, yellow and blue denote the high, medium, and low expression level of proteins respectively.

**Figure 6 ijms-21-06116-f006:**
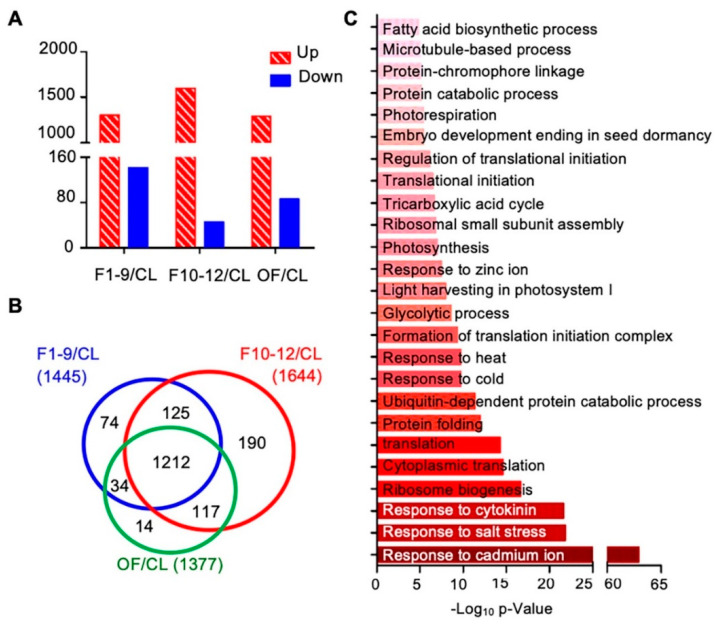
Flower-leaf differentially expressed proteins and their potential involvement in functional processes. (**A**) Numbers of upregulated and downregulated proteins in F1–9/CL, F10–12/CL, and OF/CL. (**B**) Venn diagram of the numbers of differentially expressed proteins in comparisons of stage 1–9 flowers/CL, stage 10–12 flowers/CL, and open flowers/CL. (**C**) Biological process enrichment of the overlapping 1212 proteins that who showed significant expression levels in all three floral stages. The p-value is the enrichment score (E-score) from DAVID. Higher −log10p-values indicate more significant enrichment. The dark and light red denote the higher and lower degree of GO enrichment respectively.

**Figure 7 ijms-21-06116-f007:**
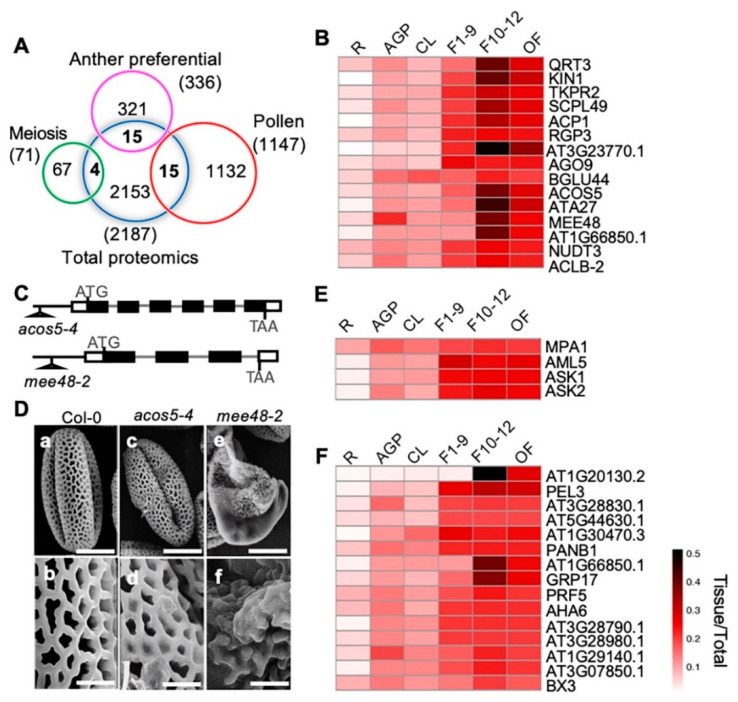
Comparison between our proteome data and published reproductive data and pollen exine phenotypic observation of mutants of two F10–12 preferentially expressed proteins. (**A**) Venn diagram shows comparison among proteomic data from this paper and previously published anther-preferential, meiosis-preferential, and pollen-related gene groups. (**B**) Heatmap shows dynamic expression of the 15 overlapping anther-preferential proteins in the six detected tissues. (**C**) The *acos5-4* and the *mee48-2* T-DNA insertion mutant alleles. (**D**) Pollen grains from Wild Type (WT) (a, b), *acos5-4* (c, d) and *mee48-2* (e, f) mutants visualized by a scanning electron microscope. Bars = 10 μm for a, c, e, 2 μm for b, d, f. (**E**) Heatmap shows dynamic expression of the four overlapping meiosis-related proteins. (**F**) Heatmap shows expression of the 15 overlapping pollen-related proteins.

**Figure 8 ijms-21-06116-f008:**
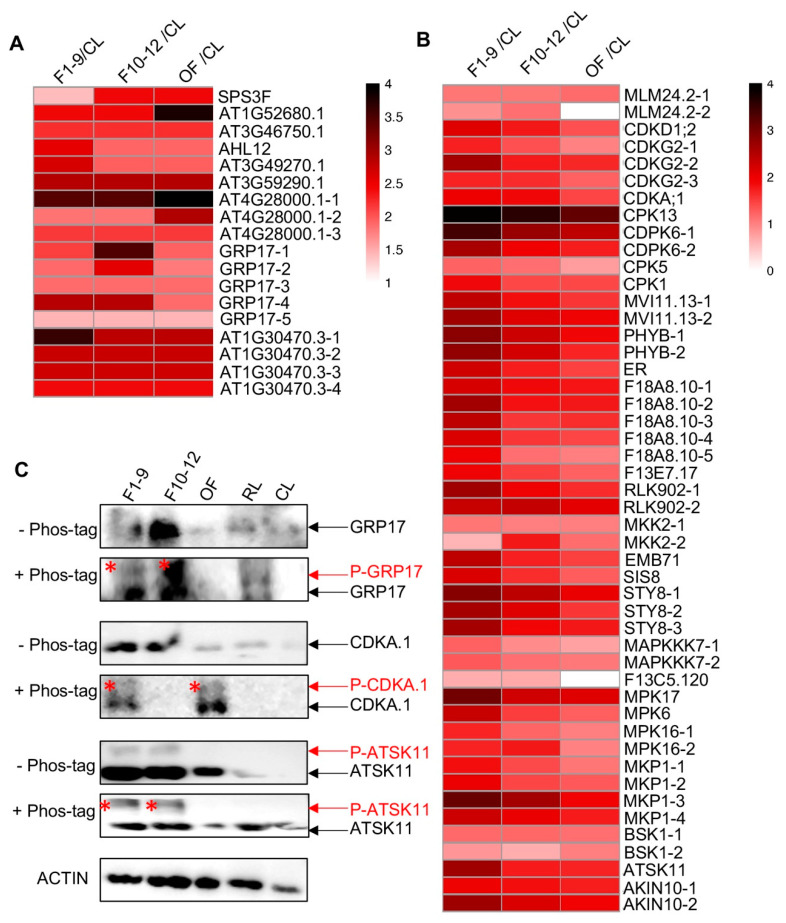
Reproductive organ-preferential phosphorylation of pollen proteins, phosphatases, and kinases. (**A**) Heat map showing the reproductive-organs-preferential-phosphorylation of pollen proteins. (**B**) Heat map showing the reproductive-organs-preferential-phosphorylation of phosphatases and kinases in reproductive organs. (**C**) Phos-tag SDS-PAGE showing the floral-organ-preferential phosphorylation of GRP17, CDC2/CDKA.1, and ATSK11 proteins in reproductive organs. The asterisk represents the phosphorylated protein.

**Figure 9 ijms-21-06116-f009:**
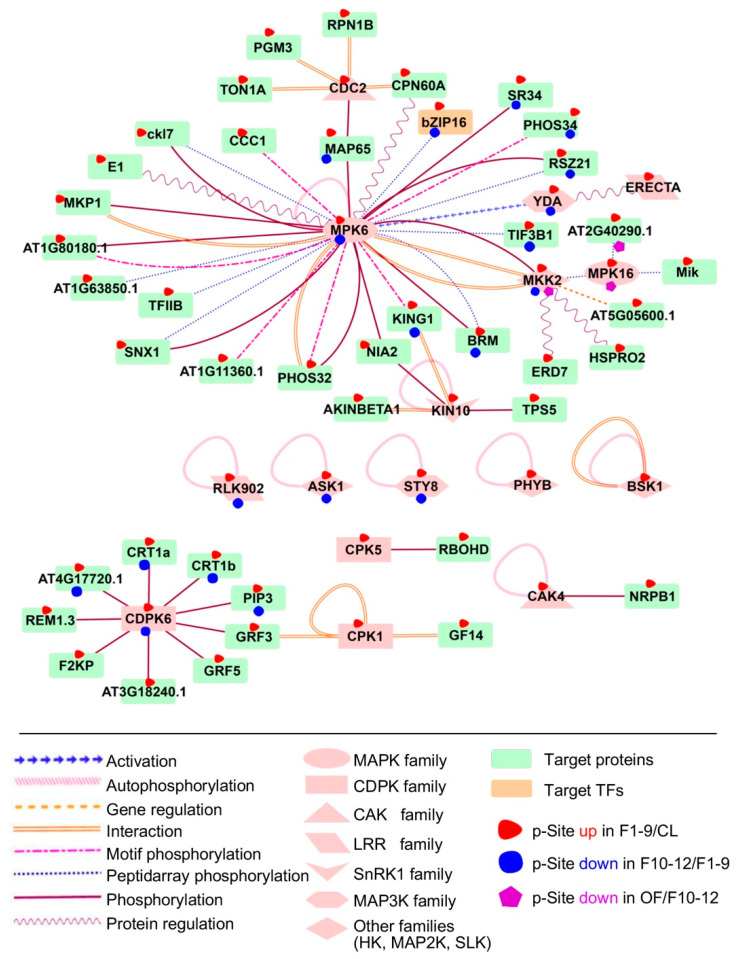
Substrates of MAPK6 and CDPK6 were highly enriched in both F1–9/CL and F10–12/F1–9 comparisons. Significant phosphorylation changement of each protein in each comparison pairs are highlighted with different coloured-shapes as indicated in legends under the illustration.

**Table 1 ijms-21-06116-t001:** Comparison of different phosphor-peptides belonging to anther-, meiosis-, and pollen-related proteins in different organ/stages.

Protein	Peptide (p-Site)	F1–9/CL	F10–12/F1–9	OF/F10–12	TAIR(The Arabidopsis Information Resource) Description
PAB5	LASDLALApSPDK	1.58/0.02	1.94/0.00	−1.24/0.01	Encodes a Class I polyA-binding protein
AML5	NMDLLDpSQLSDDDGRER	1.46/-	−1.00/-	0.00/-	A member of mei2-like gene family
SCC3	LCAEMFNpTpSDEpTDEEDENK	2.82/0.01	0.00/0.14	0.00/0.02	Essential to the monopolar orientation of the kinetochores during meiosis.
ATM	SLAPDpSPEVGR	2.31/0.04	0.00/0.07	0.00/0.04	Encodes a homolog of the human ATM gene
SPS3F	NLpSNLEIWSDDKK	1.4/0.14	1.01/0.08	0/0.05	Encodes a protein with putative sucrose-phosphate synthase activity
AT1G52680.1	NTLGMSPATNSPSSPAGpTTR	2.41/0.02	0/0.01	1.37/0.02	late embryogenesis abundant protein-related
AT3G46750.1	EVGYASLpSPR	2.2/-	0/-	0/-	low-temperature-induced protein
AHL12	SRDpSSPMSDPNEPK	2.55/-	−0.64/-	0/-	A/T (AT) hook motif DNA-binding family protein
AT3G49270.1	LpSPPPPR	2.66/-	−0.72/-	0/-	extensin-like protein
AT3G59290.1	DSGAPADDHpSQDGR	2.86/0.03	0/0.01	0/0.02	The Epsin N-Terminal Homology/ Vps27 (Vacuolar Protein Sorting), Hrs (Hepatocyte growth factor-regulated tyrosine kinase substrate) and STAM (Signal Transducing Adaptor Molecule) (ENTH/VHS) family protein
AT4G28000.1	(AT4G28000.1-1) HpTpSGNDLHSR	3.45/0.01	0/0.04	1.49/0.01	P-loop containing nucleoside triphosphate hydrolases superfamily protein
	(AT4G28000.1-2) RSIpSELTMDK	1.83/-	0/-	1.08/-	
	(AT4G28000.1-3) RNApSAASDMSSISSR	2.15/-	0/-	0/-	
GRP17	(GRP17-1) CMSGGMpSGSEGGMSR	2.12/0.00	1.37/0.03	−1.56/0.01	encodes a glycine-rich protein and is expressed specifically during flower stages 10 to 12
	(GRP17-2) SEGGISGGGMSGGSGpSK	1.89/-	0.67/-	−0.77/-	
	(GRP17-3) GMSGGSESEEGMSGSEGGMpSGGGGSK	1.87/-	0/-	0/-	
	(GRP17-4) GMSGGSEpSEEGMSGSEGGMSGGGGSK	2.84/-	0/-	−0.97/-	
	(GRP17-5) GMSGGMSGSEEGMSGpSEGGMSSGGGSK	1.45/0.02	0/0.10	0/0.04	
AT1G30470.3	(AT1G30470.3-1) SSEPEpSPHGTK	3.63/0.00	−0.8/0.00	0/0.00	SIT4 phosphatase-associated family protein
	(AT1G30470.3-2) SRDpSDDDDYHDR	2.74/0.00	0/0.01	0/0.01	
	(AT1G30470.3-3) ASGIEPTESpSPK	2.7/0.00	0/0.04	0/0.01	
	(AT1G30470.3-4) LPDESGVEPTENpSPK	2.4/-	0/-	0/-	

Note: the value is Log2fold change/P-value. F1–9 means stage 1–9 flower buds, F10–12 means stage 10–12 floral buds, OF means opened flowers, and CL means cauline leaves. “-” indicates that the peptide was only detected in one or two of the three replicates.

**Table 2 ijms-21-06116-t002:** Kinases and phosphatases that showed different expression levels in floral stages and cauline leaves.

Family	Accession	Protein Name	F1–9/CL	F10–12/CL	OF/CL	TAIR Description
CDPK	AT4G23650.1	CDPK6	1.73/0.03	1.46/0.00	1.24/0.02	Encodes calcium dependent protein kinase 3
LRR_3	AT3G17840.1	RLK902	1.00/0.07	0.99/0.02	0.61/0.06	Encodes a receptor-like kinase
LRR_3	AT1G48480.1	RKL1	1.35/0.08	1.11/0.20	0.87/0.23	receptor-like protein kinase
NDPK	AT5G63310.1	NDPK2	−0.21/0.05	0.15/0.11	−0.01/0.93	Plays a role in response to oxidative stress and UV
SnRK2	AT1G60940.1	SNRK2–10	0.65/0.04	0.64/0.08	0.47/0.15	Encodes a member of SNF1-related protein kinases

Note: The value means Log2fold change/P-value. The fold change represents the ratio of the protein expression level in a flower development stage to its expression level in cauline leaves.

**Table 3 ijms-21-06116-t003:** Comparison of different phosphor-peptides belonging to the kinases and phosphatases in different floral stages and the cauline leaves.

Family	Accession	Protein	PepTide (p-Site)	Location	F1–9/CL	F10–12/CL	OF/CL	TAIR Description
AGC	AT3G45780.1	PHOT1	ALpSESTNLHPFMTK	PAS	0/-	−0.88/-	−1.11/-	Blue-light photoreceptor.
AGC	AT3G23310.1	MLM24.2	(MLM24.2-1) MLAYpSTVGTPDYIAPEVLLK	S_TKc	1.09/0.07	1.09/0.08	1.17/0.00	Blue-light photoreceptor.
			(MLM24.2-2) DFVVAHNLSGALQpSDGRPVAPR	S_TKc	0.89/-	1.14/-	0/-	
CDK	AT1G66750.1	CAK4	ASEQNQHGNpSPAVLpSPPGK	S_TKc	2.13/-	1.78/-	1.37/-	Encodes a CDK-activating kinase that interacts with SPT5
CDK	AT1G67580.1	CDKG2	(CDKG2-1) WAAGNpSpSPTDEVEIVEEVGEK	—	1.71/0.04	1.37/0.08	0.99/0.15	Protein kinase superfamily protein;(source:Araport11)
			(CDKG2-2) MVKpSPDPLEEQR	S_TKc	2.63/0.05	1.76/0.06	1.66/0.09	
			(CDKG2-3) EGYRpSpSDpSDERGHHSLPGSR	—	1.69/-	1.64/-	1.25/-	
CDK	AT3G48750.1	CDC2	pTFpTHEVVTLWYR	S_TKc	2.01/0.01	1.88/0.01	1.45/0.01	A-type cyclin-dependent kinase.
CDPK	AT3G51850.1	CPK13	SNYpSGHDHAR	—	3.93/-	3.6/-	3.17/-	Member of calcium dependent protein kinase
CDPK	AT4G23650.1	CDPK6	(CDPK6-1) RGpSSGSGTVGSSGSGTGGSR	—	3.4/0.00	2.73/0.00	2.41/0.00	Encodes calcium dependent protein kinase 3
			(CDPK6-2) RGSpSGSGTVGSSGSGTGGSR	—	2.59/0.00	1.97/0.00	1.72/0.00	
CDPK	AT4G35310.1	CPK5	NSLNIpSMR	—	1.22/0.04	1.13/0.00	0.79/0.01	calmodulin-domain protein kinase CDPK isoform 5 (CPK5)
CDPK	AT5G04870.1	CPK1	RVSpSAGLR	S_TKc	1.87/0.01	1.43/0.00	1.43/0.01	A calcium-dependent protein kinase
CPKRK	AT3G19100.1	TAGK2	(MVI11.13-1) AFHPPpSPAR	—	2.39/0.05	1.83/0.11	1.57/0.02	Regulates gibberellic acid (GA) signaling
			(MVI11.13-2) TEpSGIFR	S_TKc	2.66/0.00	2.11/0.00	1.93/0.00	
HK	AT2G18790.1	PHYB	(PHYB -1) VpSGVGGSGGGR	—	2.84/0.07	2.29/0.02	1.91/0.03	Red/far-red photoreceptor involved in the regulation of de-etiolation.
			(PHYB -2) GGEEEPSSpSHTPNNR	—	2.77/-	2.24/-	1.68/-	
LRR_11	AT2G26330.1	ER	FGQVIpSQNSE	—	2.26/-	1.72/-	1.47/-	Homologous to receptor protein kinases
LRR_3	AT2G26730.1	F18A8.10	(F18A8.10-1) GSEGQTPPGESRpTPPR	—	2.19/0.00	1.86/0.00	1.79/0.00	Leucine-rich repeat protein kinase family protein
			(F18A8.10-2) GSEGQTPPGEpSRTPPRSVTP	—	2.65/0.00	1.81/0.00	1.78/0.00	
			(F18A8.10-3) GSEGQpTPPGESR	—	2.41/-	1.56/-	1.61/-	
			(F18A8.10-4) GSEGQTPPGESRTPPRpSVTP	—	2.18/-	1.56/-	1.44/-	
			(F18A8.10-5) GSEGQTPPGESRpTPPR	—	1.97/-	1.13/-	1.03/-	
LRR_3	AT3G02880.1	KIN7	LIEEVSHpSpSGpSPNPVSD	—	1.93/-	1.5/-	1.25/-	Leucine-rich repeat protein kinase family protein
LRR_3	AT3G17840.1	RLK902	(RLK902-1) SYVNEYpSPSAVK	—	2.66/0.00	1.98/0.00	1.63/0.00	Encodes a receptor-like kinase
			(RLK902-2) VFDLEDLLRApSAEVLGK	—	2.34/0.01	2.37/0.03	2.08/0.02	
MAP2K	AT4G29810.1	MKK2	(MKK2-1) IISQLEPEVLpSPIKPADDQLSLSDLDM	—	1.09/-	1.03/-	1.02/-	Encodes a MAP kinase kinase 2 that regulates MPK6
			(MKK2-2) FLTQSGpTFK	—	0.6/-	1.78/-	1.15/-	
MAP3K	AT1G63700.1	EMB71	AEATVpSPGSR	S_TKc	2.42/0.04	1.71/0.06	1.49/0.02	Member of MEKK subfamily
MAP3K	AT1G73660.1	SIS8	SIDVpSSSSSPR	—	2.18/0.00	1.61/0.00	1.27/0.00	Encodes a protein with similarity to MAPKKKs
MAP3K	AT2G17700.1	STY8	(STY8-1) VQIESGVMpTAETGTYR	—	2.86/-	2.45/-	2.03/-	ACT-like protein tyrosine kinase family protein
			(STY8-2) AVVASPpSQENPR	—	2.61/0.00	2.13/0.07	1.55/0.03	
			(STY8-3) VQIEpSGVMpTAETGTYR	—	2.63/-	1.91/-	1.77/-	
MAP3K	AT3G13530.1	MAPKKK7	(MAPKKK7-1) ELSIPVDQTSHpSFGR	—	1.25/0.11	0.92/0.05	0.74/0.21	Required for pollen development
			(MAPKKK7-2) VSEGKPNEASTSMPTSNVNQGDpSPVADGGK	—	1.3/-	1.14/-	1.07/-	
MAP3K	AT4G18950.1	BHP	IPEPpSVHSEEEVFEDGEEIDGGVR	—	0.67/0.02	0.7/0.02	0/0.02	Involved in mediating blue-light dependent stomatal opening
MAPK	AT2G01450.1	MPK17	LEEHNDDEEEHNpSPPHQR	—	3.04/0.00	2.26/0.00	2.14/0.00	MPK17 Map kinase family member
MAPK	AT2G43790.1	MPK6	VTSESDFMpTEpYVVTR	S_TKc	2.34/-	1.51/-	1.26/-	Encodes a MAP kinase
MAPK	AT5G19010.1	MPK16	(MPK16-1) QQHApSLPR	—	1.68/0.03	1.21/0.02	1.03/0.02	Member of MAP kinase
			(MPK16-2) VAFNDTPTAIFWpTDpYVATR	S_TKc	1.7/-	1.78/-	1/-	
Phosphatase	AT3G55270.1	MKP1	(MKP1-1) FSSLSLLPSQTpSPK	—	1.83/0.00	1.41/0.00	1.09/0.01	Encodes MAP kinase phosphatase 1
			(MKP1-2) SApSWSASR	—	2.02/0.05	1.44/0.04	1.3/0.03	
			(MKP1-3) GVNTFLQPpSPNRK	—	3.1/-	2.64/-	1.95/-	
			(MKP1-4) pSLDEWPK	—	2.29/-	2.02/-	1.74/-	
RLCK_2	AT4G35230.1	BSK1	(BSK1-1) SYpSpTNLAYTPPEYLR	—	1.18/-	1.19/-	1.13/-	Encodes Brassinosteroid (BR)-signaling kinase 1
			(BSK1-2) KQEEAPpSpTPQRPLSPLGEACSR	—	0.84/-	0.67/-	1.03/-	
SLK	AT5G26751.1	SK 11	GEPNIpSpYICSR	S_TKc	2.66/0.01	1.75/0.00	1.69/0.00	Encodes a SHAGGY-related kinase involved in meristem organization.
SnRK1	AT3G01090.1	KIN10	(AKIN10-1) pTSCGSPNYAAPEVISGK	S_TKc	1.96/0.00	1.84/0.01	1.72/0.00	Encodes a SNF1-related protein kinase
			(AKIN10-2) MHPAEpSVASPVSHR	—	2.68/0.01	2.18/0.04	1.94/0.05	

Note: Values mean Log2fold change/P-value. The fold change represents the ratio of the phosphorylation level of the detected peptide in a flower development stage to its phosphorylation level in cauline leaves. F1–9 means stage 1–9 flower buds, F10–12 means stage 10–12 floral buds, OF means opened flowers, and CL means cauline leaves. The domains in which each peptide is located are shown. PAS indicates the Per-ARNT-Sim (PAS) domain; S-TKc indicates the serine/threonine-protein-kinases catalytic domain; “—” indicates that the peptide was not in a specific domain.

**Table 4 ijms-21-06116-t004:** Phosphorylation sites of kinases and phosphatases showing different phosphorylation levels in the comparison of F10–12 to F1–9 and OF to F10–12.

Family	Accession	Protein	Peptide (p-Site)	Location	F10–12/F1–9	OF/F10–12	TAIR Description
CDK	AT1G67580.1	CDKG2	MVKpSPDPLEEQR	—	−0.87/0.04	0/0.14	Protein kinase superfamily protein
CDPK	AT4G23650.1	CDPK6	(CDPK6_1) RGpSSGSGTVGSSGSGTGGSR	—	−0.67/-	0/-	Encodes calcium dependent protein kinase 3
			(CDPK6_2) RGSpSGSGTVGSSGSGTGGSR	—	−0.63/0.01	0/0.04	
LRR_3	AT2G26730.1	F18A8.10	(F18A8.10_2) GSEGQTPPGEpSRTPPRSVTP	—	−0.84/-	0/-	Leucine-rich repeat protein kinase family protein
			(F18A8.10_3) GSEGQpTPPGESR	—	−0.86/-	0/-	
			(F18A8.10_5) GSEGQTPPGESRpTPPR	—	−0.85/0.07	0/0.50	
LRR_3	AT3G17840.1	RLK902	SYVNEYpSPSAVK	—	−0.69/0.01	0/0.05	Encodes a receptor-like kinase
MAP2K	AT4G29810.1	MKK2	FLTQSGpTFK	—	0	−0.62/-	Encodes a MAP kinase kinase 2 that regulates MPK6 and MPK4
MAP3K	AT1G63700.1	YDA	AEATVpSPGSR	—	−0.71/0.15	0/0.21	Member of MEKK subfamily
MAP3K	AT2G17700.1	STY8	VQIEpSGVMpTAETGTYR	—	−0.72/0.06	0/0.02	ACT-like protein tyrosine kinase family protein
MAPK	AT2G01450.1	MPK17	LEEHNDDEEEHNpSPPHQR	—	−0.78/0.01	0/0.24	MPK17 Map kinase family member
MAPK	AT2G43790.1	MPK6	VTSESDFMpTEpYVVTR	S_TKc	−0.82/-	0/-	Encodes a MAP kinase
MAPK	AT5G19010.1	MPK16	VAFNDTPTAIFWpTDpYVATR	S_TKc	0/-	−0.78/-	Member of MAP Kinase
phosphatase	AT3G55270.1	MKP1	GVNTFLQPpSPNRK	—	0/-	−0.69/-	Encodes MAP kinase phosphatase 1
SLK	AT5G26751.1	SK 11	GEPNIpSpYICSR	S_TKc	−0.91/0.04	0/0.31	Encodes a SHAGGY-related kinase involved in meristem organization

Note: Values in each stage mean Log2fold change/P-value. The fold change represents the ratio of phosphorylation level of the detected peptide in a certain flower development stage to its phosphorylation level in another flower development stage. F1–9 means stage 1–9 flower buds, F10–12 means stage 10–12 floral buds, OF means opened flowers, and CL means cauline leaves. The domains in which each peptide is located are shown. PAS indicates the Per-ARNT-Sim (PAS) domain; S-TKc indicates the serine/threonine-protein-kinases catalytic domain; “—” indicates that the peptide was not in a specific domain.

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
