# Peer review of "Global Quantitative Proteomics Studies Revealed Tissue-Preferential Expression and Phosphorylation of Regulatory Proteins in Arabidopsis"

_ijms, 2020, doi:10.3390/ijms21176116_

Round 1
Reviewer 1 Report
The study reports on the proteomic and phosphoproteomic analysis of six stages/organs of Arabidopsis thaliana plants. Differences in protein level and phosphorylation are discussed, suggesting specific protein functions and regulatory networks during plant development.
The methods are well described and the quality of proteomic and phosphoproteomic data is good, although some bias in the quantification may be due to the iTRAQ method, as the authors point out in the discussion.
I think the work is suitable for publication, however improvement of text and Table is required.
Main corrections:
1) The quality of the Tables 1-4 should be improved. The text should be formatted in order to be better readable. The short TAIR description of the proteins should be added as additional column.
2) Lines 304-312: authors say that they "chose two proteins, ACOS5 and MEE48, that showed the highest expression level in F10-12 for functional test using mutations". However the role of these two proteins had already been described in previous works. The ACOS5 gene has already been studied for its role in pollen development:
"ACOS5 is required for primexine formation and exine pattern formation during microsporogenesis in Arabidopsis"
Journal of Plant Biology volume 60, pages404–412(2017)
Also for MEE48 a previously published work reported defects in pollen formation in a mutant line: "A Large-Scale Genetic Screen in Arabidopsis to Identify Genes Involved in Pollen Exine Production" Plant physiology 157(2):947-70, DOI: 10.1104/pp.111.179523
The functional test on ACOS5 and MEE48 mutant lines should refer to the previously published works.
3) Although the work is well written, there are a number of typing or grammatical errors that should be corrected after careful proofreading.
Here are just a few examples:
Line 282: "as anther" should be "as anther"
Line 304 "The highly expression levels" should be "The high expression levels"
Line 305 " we choose two proteins" should be "we chose two proteins"
Author Response
Point 1: The quality of the Tables 1-4 should be improved. The text should be formatted in order to be better readable. The short TAIR description of the proteins should be added as additional column.
Response 1: The quality of the Tables 1-4 has been improved in the revised manuscript. In order to be better readable, a column showing the short TAIR description of each protein has been added to Tables 1-4.
Point 2: Lines 304-312: authors say that they "chose two proteins, ACOS5 and MEE48, that showed the highest expression level in F10-12 for functional test using mutations". However, the role of these two proteins had already been described in previous works. The ACOS5 gene has already been studied for its role in pollen development: “ACOS5 is required for primexine formation and exine pattern formation during microsporogenesis in Arabidopsis" Journal of Plant Biology volume 60, pages404–412(2017); Also for MEE48 a previously published work reported defects in pollen formation in a mutant line: "A Large-Scale Genetic Screen in Arabidopsis to Identify Genes Involved in Pollen Exine Production" Plant physiology 157(2):947-70, DOI: 10.1104/pp.111.179523. The functional test on ACOS5 and MEE48 mutant lines should refer to the previously published works.
Response 2: Thanks for this comment. Published papers related to ACOS5 and MEE48 have been referred to the functional analyses on the novel acos5-4 and mee48-2 alleles. In order to descript more clearly, we added the following texts in line 312-318, page 12 of the revised manuscript:
“Previous studies base on phenotypic analyses suggests that QRT3 plays a direct role in degrading the pollen mother cell wall during microspore development (Rhee et al., 2003), ACOS5 participates in a conserved biochemical pathway of sporopollenin monomer biosynthesis and is required for primexine formation (de Azevedo Souza et al., 2009; Xie et al., 2017), and MEE48/A6 is important for the dissolution of callose wall around the microspore tetrads (Hird et al., 1993; Dobritsa et al., 2011).”
In addition, we revised the description of about acos5-4 and mee48-2 alleles as follows:
“In addition, our observation on another two T-DNA insertional mutant alleles, one ACOS5 allele acos5-4, and one MEE48 allele mee48-2 (Figure 7C) also showed that, unlike the wild type plants that produced normal pollen grains with well-organized pollen exine structure, the acos5-4 mutant had shriveled pollens with abnormal pollen exine structure. Pollen defects of mee48-2 were stronger and most of pollen grains were shrunken and the pollen exine components were not even connected (Figure 7D).”
Point 3: Although the work is well written, there are a number of typing or grammatical errors that should be corrected after careful proofreading.
Here are just a few examples:
Line 282: "as anther" should be "as anther"
Line 304 "The highly expression levels" should be "The high expression levels"
Line 305 " we choose two proteins" should be "we chose two proteins"
Response 3: All above mentioned errors in writing have been corrected in the revised manuscript.
Reviewer 2 Report
The manuscript presented by Jianan Lu and colleagues provides extensive data on protein expression and phosphorylation in Arabidopsis. These results will undoubtedly be useful for subsequent work with individual protein networks that regulate organogenesis.
The authors used the high-resolution mass spectrometry, offered by LTQ-Orbitrap Elite mass spectrometer. In my opinion, the technical part of the job is well done. An interesting part is post-translational modifications, in which phosphorylation sites have been found for several important proteins. One of the largest challenges with PTMs tends to be the localization of the modification on the peptide backbone. The authors focused on phosphorylation; this PTM is low abundant compared to the abundance of the precursor, and therefore it would be useful to write in more detail the methodology regarding effectiveness of enrichment and stabilization of the modifications (especially TiO2 enrichment). In general, because the authors used HRAM spectroscopy, the data seems to be reliable. The section 4.7 Bioinformatics analysis is described too briefly. From the article, I also did not understand if the mass spectrometry data have been deposited to ProteomeXchange via the PRIDE partner repository.
In this regard, and referring to the general discussion about the role of this kind of global experiments, it would be appropriate to say a few words about the problem as a whole. Discussions in such articles are often weak; we see the same situation in this article. The authors describe many interesting findings in the results (links with hormonal signaling and ROS, the phosphorylation level in CMSGGMpSGSEGGMSR peptide of GRP17, the data on At4G28000.1 and AT1G52680.1, KIN1 expression, etc.), but do not discuss them enough. I do not insist on expanding the discussion, the work is already large, but it is clear that it requires continuation and further decoding. In particular, visualization of networks would be very helpful. The phosphorylation regulatory network is presented in Dataset 6 (Supplemental information) using Cytoscape 3.4.0 for visualization. However, this file cannot be read on my computer. I suggest moving this picture into the main text.
I would also like to see some idea about the different expression of defense-related proteins in different organs.
Considering the «threshold» discussion, I agree with the authors that they use the threshold value 1.5, but not 1.2. Perhaps threshold 2 would be even better, as priority should be given to the reliability of the results.
A few small suggestions:
Please use the term “phosphopeptides” and “phosphoproteins” instead of “phosphor-proteins” throughout the text.
Line 233. The comparison of results
- 338-341. The same sentence is repeated twice.
- 447. “families, CDPK6, RLK1, RLK902, and SNF1 were up-regulated in floral stages (Table 2).” –Please check all protein names. For instance, SNF1 is SNF1-related protein kinase catalytic subunit alpha KIN11 (the common abbreviation is KIN11). https://www.arabidopsis.org/servlets/TairObject?type=locus&name=AT3G29160
https://www.uniprot.org/uniprot/P92958
https://thebiogrid.org/7895/summary/arabidopsis-thaliana/kin11.html
- 617. Thermo Fisher Scientific
- 686 Int. J. Dev. Biol. Please check all references carefully.
Author Response
Point 1: The authors used the high-resolution mass spectrometry, offered by LTQ-Orbitrap Elite mass spectrometer. In my opinion, the technical part of the job is well done. An interesting part is post-translational modifications, in which phosphorylation sites have been found for several important proteins. One of the largest challenges with PTMs tends to be the localization of the modification on the peptide backbone. The authors focused on phosphorylation; this PTM is low abundant compared to the abundance of the precursor, and therefore it would be useful to write in more detail the methodology regarding effectiveness of enrichment and stabilization of the modifications (especially TiO2enrichment). In general, because the authors used HRAM spectroscopy, the data seems to be reliable.
Response 1: Thanks. The methodology regarding effectiveness of enrichment and stabilization of the modifications (especially TiO2 enrichment) has been revised to show details, including the Phosphopeptide enrichment using TiO2 microcolumns in section 4.6 and methodology of phosphoproteomic data analysis in section 4.7 (line 703-722, page 27):
“4.6. Phosphopeptide enrichment
In addition to total proteome quantification, most of the iTRAQ–labeled peptide mixtures (400 ug for each) were used for phosphopeptide enrichment using TiO2 microcolumns (1350L250W046 Titansphere, 5 mm, 250 4.6 mm, GL sciences Inc) as described by Thingholm et al.(Thingholm et al., 2006). The mixtures were vacuum dried and resuspended with TiO2 loading buffer (1 M glycolic acid in 80% ACN, 5% TFA) and applied onto the TiO2 microcolumn. After washing four times with 20 μl loading buffer and at least three times with 20 μl washing buffer (80% ACN, 1% TFA), the bound peptides were eluted twice with 20 μl elution buffer 1 (2 M NH3•H2O), and with 2 μl elution buffer 2 (1 M NH3•H2O in 30% ACN). The eluates from TiO2 enrichment were desalted as described in the total proteome.
4.7 Phosphoproteomic data analysis
Like the proteome data, raw LC-MS/MS files from phoshoproteomics experiments were queried against the Arabidopsis database using an in-house processed using Mascot search engine 2.3.02 (Matrix Science, London, UK), with the same parameters except for four significant points: 1. phospho_STY (serine, threonine, and tyrosine) was added in the specified parameters in the protein database searches. 2. Identified peptides were further validated with a Target Decoy PSM validator, and phosphorylation sites were evaluated with phosphoRS3.0. 3. A false discovery rate (FDR) of 1.0% was used for the identification of phosphorylation residues (phosphosites). 4. Regarding the phosphorylation levels of phosphoproteins, we determined the ratios of each phosphosites manually instead of using the calculated results from the Proteome Discover.”
Point 2: The section 4.7 Bioinformatics analysis is described too briefly.
Response 2: The section “4.7 Bioinformatics analysis” (current 4.8) has been revised in the current paper. Methods and software used for GO analysis, pathway enrichment analysis, substrates enrichment, subcellular localization prediction, DEPs clustering, and heatmap drawing were revised as follows (line 723-732 of page 27):
“4.8. Gene ontology and pathway enrichment analyses
Gene Ontology (GO) analysis of differentially expression genes was performed using online DAVID analysis (http://david.abcc.ncifcrf.gov/). Pathway enrichment analysis was performed using the list of genes that encode for proteins containing the peptide/phosphopeptides from each of the inferred cluster. Pathway enrichment with a set of genes was evaluated by comparing that set of genes against genes within known pathways using Mapman software. The subcellular localization of proteins and phosphoproteins was predicted with WoLF PSORT (http://above-ground.genscript.com/wolf-psort.html). R language (x64 3.3.1) was used to cluster the DEPs and to draw heatmaps. The relationship of detected kinases and potential substrates was generated through PhosPhATDB (http://phosphat.uni-hohenheim.de/) (Zulawski et al., 2013).”
Point 3: From the article, I also did not understand if the mass spectrometry data have been deposited to ProteomeXchange via the PRIDE partner repository.
Response 3: The original proteomic and phospho-proteomic data from this article has been deposited to the ProteomeXchange Consortium with the dataset identifier PXD019691. The above sentence is provided in line 737-738, page 28 of the revised paper.
Point 4: In this regard, and referring to the general discussion about the role of this kind of global experiments, it would be appropriate to say a few words about the problem as a whole. Discussions in such articles are often weak; we see the same situation in this article. The authors describe many interesting findings in the results (links with hormonal signaling and ROS, the phosphorylation level in CMSGGMpSGSEGGMSR peptide of GRP17, the data on At4G28000.1 and AT1G52680.1, KIN1 expression, etc.), but do not discuss them enough. I do not insist on expanding the discussion, the work is already large, but it is clear that it requires continuation and further decoding.
Response 4: Thanks for your understanding and for this comment. We agree that expanding discussion on the findings would be better. However, as a result of the huge proteomic/phosphoproteomic data and experimental validation of selected proteins/PTMs in this paper, as well as the lack of further functional studies for most of interesting proteins and modifications, we did not discuss these findings in depth separately in the discussion section. Nevertheless, we did briefly discuss following the description of each protein/PTM, such as: “suggesting potentially important function of this phosphorylation in flower and pollen development.” for At4G28000 in line 444-446, and “suggesting a possible role for phosphorylation of this protein in flower developmental stages especially in the pollen” for AT1G52680.1 in line 458-459 of page 15.
Point 5: In particular, visualization of networks would be very helpful. The phosphorylation regulatory network is presented in Dataset 6 (Supplemental information) using Cytoscape 3.4.0 for visualization. However, this file cannot be read on my computer. I suggest moving this picture into the main text.
Response 5: This picture has been moved to the main text as Figure 9, and the figure legend has been added, as follows:
“Figure 9. Substrates of MAPK6 and CDPK6 are highly enriched in both F1-9/CL and F10-12/F1-9 comparisons. Significant phosphorylation changement of each protein in each comparison pairs are highlighted with different coloured-shapes as indicated in legends under the illustration.”
Point 6: I would also like to see some idea about the different expression of defense-related proteins in different organs.
Response 6: The 275 proteins that show clearly sage/organ-preferential expression pattern (Figure S3B-H) were searched with keyword “defense” and 9 defense-related proteins were found to be differentially expressed in different organs/stages. These 9 proteins include ESM1, CA2, PSBP-1, ESP, TSA1, MBP1, MLP28, MLP423, and F9D16.150. Description of the results has been added in line 339-359, page 13 of the revised paper as follows:
2.6. different expression of defense-related proteins in different organs
To identify the organ/stage-preferential expression of defense proteins, the 275 proteins that show clearly sage/organ-preferential expression pattern (Figure S3B-H) were searched with keyword “defense”. Nine defense related proteins were identified (Figure S5 and Dataset 3). Among them, three proteins showed strong cauline leaf-preferential expression. These 3 proteins include EPITHIOSPECIFIER MODIFIER 1 (ESM1) that mediates indol-3-acetonitrile production from indol-3-ylmethyl glucosinolate for defenses against insect herbivores and various pathogens (Burow et al., 2008), CARBONIC ANHYDRASE 2 (CA2) that is structurally required for the assembly of Summary Complex I mitochondrial electron transport chain (mETC) and plays important role in reproductive development (Fromm et al., 2016),and PHOTOSYSTEM II SUBUNIT P-1 (PSBP-1) that participates in the regulation of oxygen evolution and is involved in defense response to temperature and bacterium (Gargano et al., 2013; Thomas et al., 2013).
On the other hand, compared to their expression in vagetative tissues, the expression of 6 proteins were clearly higher in the three flower developmental stages, especially F10-12 (Figure S5 and Dataset 3). These 6 proteins are EPITHIOSPECIFIER PROTEIN (ESP), TSK-ASSOCIATING PROTEIN 1 (TSA1), MYROSINASE-BINDING PROTEIN 1 (MBP1), MLP-LIKE PROTEIN 28 (MLP28), MLP-LIKE PROTEIN 423 (MLP423), and F9D16.150 (Figure S5 and Dataset 3). Among these proteins, ESP is suggested to be involved in pathogen resistance and leaf senescence (Buxdorf et al., 2013; Koyama et al., 2013; Backenkohler et al., 2018); TSA1 is suggested to be Jasmonic acid (JA) inducible and facilitates ER body formation (Geem et al., 2019), be involved in nuclear architecture (Batzenschlager et al., 2013) and seedling development in darkness (Li et al., 2011).
Point 7: Considering the «threshold» discussion, I agree with the authors that they use the threshold value 1.5, but not 1.2. Perhaps threshold 2 would be even better, as priority should be given to the reliability of the results.
Response 7: Thanks. We agree that threshold 2 might enhance the reliability of detected variants, although it might also overlook useful information with small changes. In our opinion, the key issue of selection of candidates for biological studies does not only rest in 1.2, 1.5, or 2.0 fold of alteration of peptide amount, but also the confidence level of the selections and how willing the scientist is to spend extra effort to validate the selected candidates. Therefore, to explain our opinion clearly, we added descriptions in the discussion section (line 617-621, page 25) as follows:
“Another question is how the threshold value of differentially expressed proteins can be standardized, and when is appropriate to use 1.2, 1.5, or 2. The use of lower ratio such as 1.2 would include more quantification variations; however, there would be more false positive variations. In comparison, higher ratio such as 2 might enhance the reliability of detected variations, but it also possibly overlook useful information with small changes. Therefore, the key issue of selection of candidates should base on two aspects, one is the confidence level of the quantitative data, the correlation of biological triplicate for example, and the other is how willing the scientist is to spend extra effort to validate the selected candidates with an alternative method, including immunoblot, multiple reaction monitoring (MRM), or selected reaction monitoring (SRM) of samples.”
Point 8: A few small suggestions:
Please use the term “phosphopeptides” and “phosphoproteins” instead of “phosphor-proteins” throughout the text.
Response 8: “phosphor-proteins” and “phosphor-peptides” have been replaced with “phosphoproteins” and “phosphopeptide” throughout the text.
Point 9: Line 233. The comparison of results
Response 9: This sentence has been revised as suggested.
Point 10: 338-341. The same sentence is repeated twice.
Response 10: We have deleted the repeated sentence.
Point 11: “families, CDPK6, RLK1, RLK902, and SNF1 were up-regulated in floral stages (Table 2).” –Please check all protein names. For instance, SNF1 is SNF1-related protein kinase catalytic subunit alpha KIN11 (the common abbreviation is KIN11).
Response 11: We apologize for the typo for “SNF1” in line 490. It should be SNF1-related protein kinase 2.10/2B (SnRK2-10/SRK2B) (AT1G60940.1). The name in Table 2 is correct. We have revised it accordingly and checked all protein names throughout the text.
Point 12: 617 Thermo Fisher Scientific
Response 12: This error has been corrected in the revised manuscript.
Point 13: 686 Int. J. Dev. Biol. Please check all references carefully.
Response 13: The references have been checked carefully and the format errors have been revised.
